# The importance of Antarctic krill in biogeochemical cycles

E.L. Cavan[1,12]*, A. Belcher [2], A. Atkinson [3], S.L. Hill [2], S. Kawaguchi[4],
S. McCormack [1,5], B. Meyer[6,7,8], S. Nicol[1], L. Ratnarajah[9], K. Schmidt[10],
D.K. Steinberg[11], G.A. Tarling[2] & P.W. Boyd[1,5]

Antarctic krill (*Euphausia superba*) are swarming, oceanic crustaceans, up to two inches long, and best known as prey for whales and penguins – but they have another important role. With their large size, high biomass and daily vertical migrations they transport and transform essential nutrients, stimulate primary productivity and influence the carbon sink. Antarctic krill are also fished by the Southern Ocean's largest fishery. Yet how krill fishing impacts nutrient fertilisation and the carbon sink in the Southern Ocean is poorly understood. Our synthesis shows fishery management should consider the influential biogeochemical role of both adult and larval Antarctic krill.

Ocean biogeochemical cycles are paramount in regulating atmospheric carbon dioxide ($CO_2$) levels and in governing the nutrients available for phytoplankton growth[1]. As phytoplankton are essential in most marine food webs, biogeochemistry is also important in fuelling fishery production[2]. The role of phytoplankton in atmospheric $CO_2$ drawdown and fish production has been the central focus of many biogeochemical studies (e.g., refs. [3,4]). However, despite evidence of their potential importance, higher organisms (metazoa) such as zooplankton (e.g., copepods and salps), nekton (e.g., adult krill and fish), seabirds and mammals[5–12], have received less attention concerning their roles in the global biogeochemical cycles.

One of the main mechanisms by which metazoa can influence biogeochemical cycles is through the biological pump[1] (Fig. 1). The biological pump describes a suite of biological processes that ultimately sequester atmospheric $CO_2$ into the deep ocean on long timescales. During photosynthesis in the surface, ocean phytoplankton produce organic matter and a fraction (< 40 %) sinks to deeper waters[13]. It is estimated that 5–12 Gt C is exported from the global surface ocean annually[14], with herbivorous metazoa contributing to the biological pump by releasing fast-sinking faecal pellets, respiring carbon at depth originally assimilated in the surface ocean and by excreting nutrients near the surface promoting further phytoplankton

[1]Institute for Marine and Antarctic Studies, University of Tasmania, Hobart, TAS, Australia. [2]British Antarctic Survey, Natural Environment Research Council, High Cross, Madingley Rd, Cambridge CB3 0ET, UK. [3]Plymouth Marine Laboratory, Prospect Place, The Hoe, Plymouth PL1 3DH, UK. [4]Australian Antarctic Division, Kingston, TAS, Australia. [5]Antarctic Climate and Ecosystems CRC, University of Tasmania, Hobart, TAS, Australia. [6]Alfred Wegener Institute for Polar and Marine Research, Bremerhaven, Germany. [7]Institute for Chemistry and Biology of the Marine Environment, University of Oldenburg, Carl-von-Ossietzky-Straße 9-11, 26111 Oldenburg, Germany. [8]Helmholtz Institute for Functional Marine Biodiversity at the University of Oldenburg, Ammerländer Heerstrasse 231, Oldenburg 26129, Germany. [9]Department of Earth, Ocean and Ecological Sciences, University of Liverpool, Liverpool, UK. [10]School of Geography, Earth and Environmental Science, University of Plymouth, Plymouth, UK. [11]Virginia Institute of Marine Science, College of William & Mary, Williamsburg, VA, USA. [12]Present address: Department of Life Sciences, Imperial College London, Silwood Park Campus, Buckhurst Road, Ascot, Berkshire SL5 7PY, UK. *email: e.cavan@imperial.ac.uk

 **1**

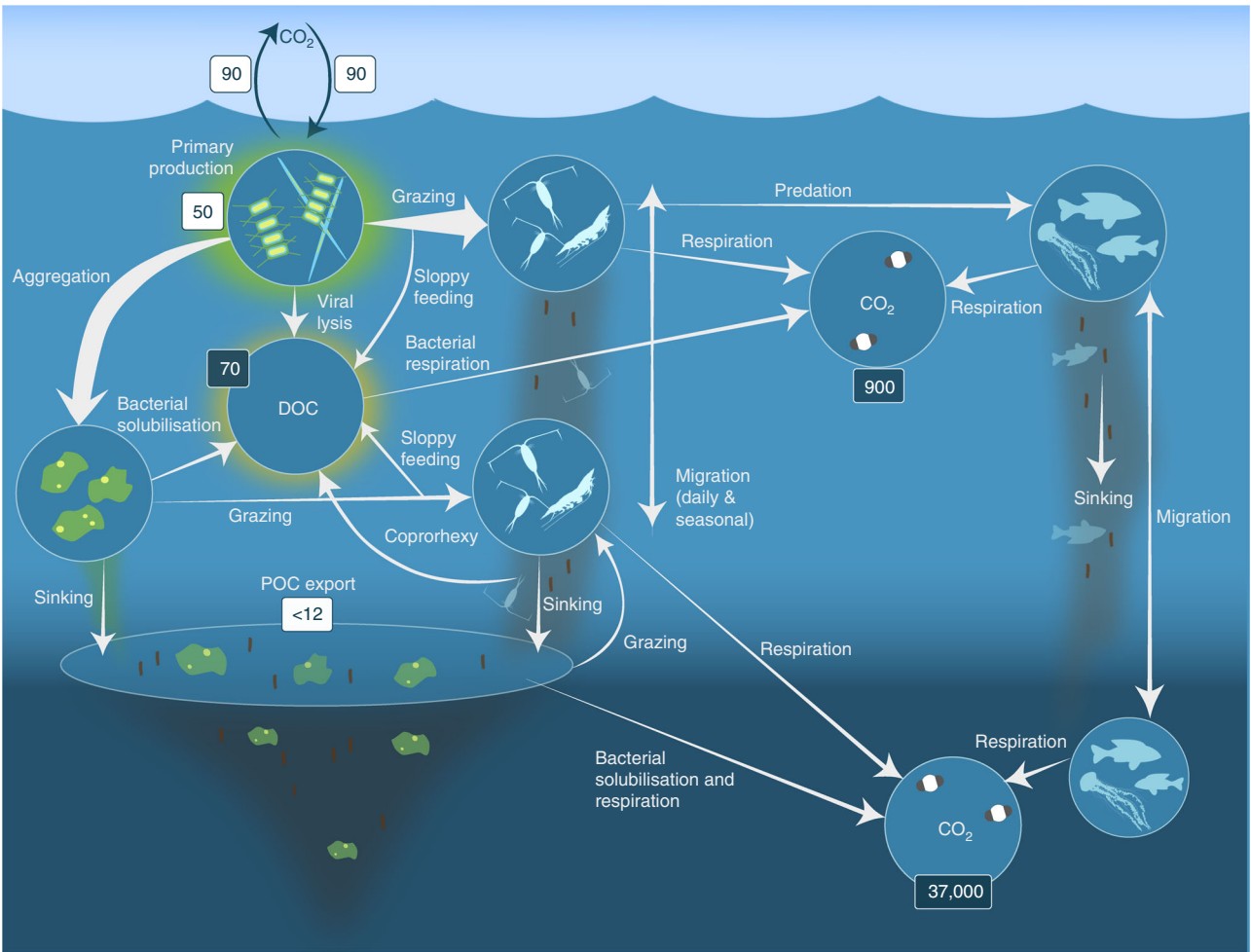

**Fig. 1** Processes in the biological pump. Phytoplankton convert $CO_2$, which has dissolved from the atmosphere into the surface oceans (90 Gt yr$^{-1}$) into particulate organic carbon (POC) during primary production (~ 50 Gt C yr$^{-1}$). Phytoplankton are then consumed by krill and small zooplankton grazers, which in turn are preyed upon by higher trophic levels. Any unconsumed phytoplankton form aggregates, and along with zooplankton faecal pellets, sink rapidly and are exported out of the mixed layer (< 12 Gt C yr$^{-1}$ [14]). Krill, zooplankton and microbes intercept phytoplankton in the surface ocean and sinking detrital particles at depth, consuming and respiring this POC to $CO_2$ (dissolved inorganic carbon, DIC), such that only a small proportion of surface-produced carbon sinks to the deep ocean (i.e., depths > 1000 m). As krill and smaller zooplankton feed, they also physically fragment particles into small, slower- or non-sinking pieces (via sloppy feeding, coprorhexy if fragmenting faeces), retarding POC export. This releases dissolved organic carbon (DOC) either directly from cells or indirectly via bacterial solubilisation (yellow circle around DOC). Bacteria can then remineralise the DOC to DIC ($CO_2$, microbial gardening). Diel vertically migrating krill, smaller zooplankton and fish can actively transport carbon to depth by consuming POC in the surface layer at night, and metabolising it at their daytime, mesopelagic residence depths. Depending on species life history, active transport may occur on a seasonal basis as well. Numbers given are carbon fluxes (Gt C yr$^{-1}$) in white boxes and carbon masses (Gt C) in dark boxes

growth[7,8]. It is essential that the role of metazoa in biogeochemical cycles is recognised[15] to improve the mechanistic understanding of the present-day environment. This will enable predictions of how biogeochemical cycles may change in the future, as metazoa become impacted by multiple anthropogenic pressures such as fishing and climate change[16,17].

Here we focus on the role of krill (specifically Antarctic krill, *Euphausia superba*) in biogeochemical cycles for three key reasons: this single species has extraordinarily high biomass (Box 1) and so could have large impacts on biogeochemical cycles[18]; they are among the largest pelagic crustaceans with commensurately high swimming speeds; they have high grazing capacity, large fast-sinking faecal pellets and the ability to migrate vertically; and the fact that little attention has been given to assessing the importance of krill in ocean biogeochemical cycles. This review is centred on *E. superba* as the Southern Ocean is the location of the largest krill fishery[19], one of the largest carbon sinks globally[20], and a site of water mass formation transporting nutrients

throughout the global oceans[21]. *E. superba* are also one of the few pelagic crustacea to be commercially harvested, as most fished crustacea are bottom-dwelling (e.g., crabs and prawns)[22]. The majority of processes discussed here are also relevant to other fisheries, particularly those of small pelagic fish such as anchovy and sardines, which are also feeders on plankton and may be important in biogeochemical cycles[23,24]. Management of the Southern Ocean *E. superba* fishery is currently centred around the importance of *E. superba* in supporting populations of megafauna (e.g., seals, penguins, whales etc.) and maintaining a sustainable commercial fishery[19]. At present, there is no consensus on the effect that harvesting large quantities of *E. superba* (or any other species) could have on global ocean biogeochemical cycles and hence atmospheric $CO_2$ levels. Here, we synthesise current knowledge on the importance of *E. superba* in regulating ocean biogeochemical cycles and consider the effects that commercial harvesting of these krill could have on ocean biogeochemistry.

---

**Box 1. | The ecology and abundance of *E superba***

*E. superba* are by far the dominant of the seven Southern Ocean krill species in terms of biomass[128]. They display a circumpolar distribution, which largely coincides with the extent of winter sea ice[129]. Typically, *E. superba* live for 5–6 years in the wild, and grow up to 65 mm in length, and hence are larger than other abundant krill species some of which (e.g., *Meganyctiphanes norvegica* and *Euphausia pacifica*) play crucial roles in northern marine ecosystems. These northern species are key members of much more diverse ecosystems, so rarely dominate the pelagic biomass the way that *E. superba* does. This important ecological role is reflected in the way *E. superba* are represented in Southern Ocean foodweb models, where they are parameterised as their own functional, species-resolved group, whereas other euphausiids are combined with other species[112]. Even though all krill are much larger than many planktonic species, they are often regarded as plankton, however the strong swimming abilities of adult krill are a characteristic feature of nekton[130]. Krill form some of the largest monospecific aggregations (swarms) in the animal kingdom[131], making them a critical food item for whales, seals and seabirds, and the target of the Southern Ocean's largest fishery. *E. superba* themselves are a major grazer of primary production in the Southern Ocean[68].

The vast spatial distribution of adult *E. superba*, 19 million km$^2$ [132], means that it is currently very difficult to conduct a synoptic survey of the entire population, resulting in highly uncertain estimates of krill biomass. The preferred method for assessing krill biomass is via hydroacoustics, but this involves methodological uncertainties and does not survey either surface (< 20 m) or deep water[107]. Best estimates from acoustics of post-larval *E. superba* density in the southwest Atlantic Southern Ocean was 29 g m$^{-2}$ (biomass = 60 million tonnes) in 2000[18], and 5.5 g m$^{-2}$ (1996)[133] and 23 g m$^{-2}$ (2006)[134] at two different Indian Ocean sector sites.

Circumpolar estimates of *E. superba* abundance are based on net data, for which there is a much longer (since the 1920s) historical record of data[135] than for acoustics. Nets also have their own limitations associated with animal avoidance, low sampling frequency and limited sampling of the water column, typically up to 200 m[107]. Data from multiple scientific net surveys over several decades estimate circumpolar post-larval biomass at 379 million tonnes, with the highest biomass located in the Southwest Atlantic[132]. However, combining net and acoustic data gives a slightly lower circumpolar biomass estimate of 215 million tonnes[132]. The high Southern Ocean biomass of *E. superba* is further confirmed by the large number of *E. superba*-dependent predators. Combined with body size, the high biomass of *E. superba* means they are likely a significant vector for recycling nutrients and transporting carbon on large scales.

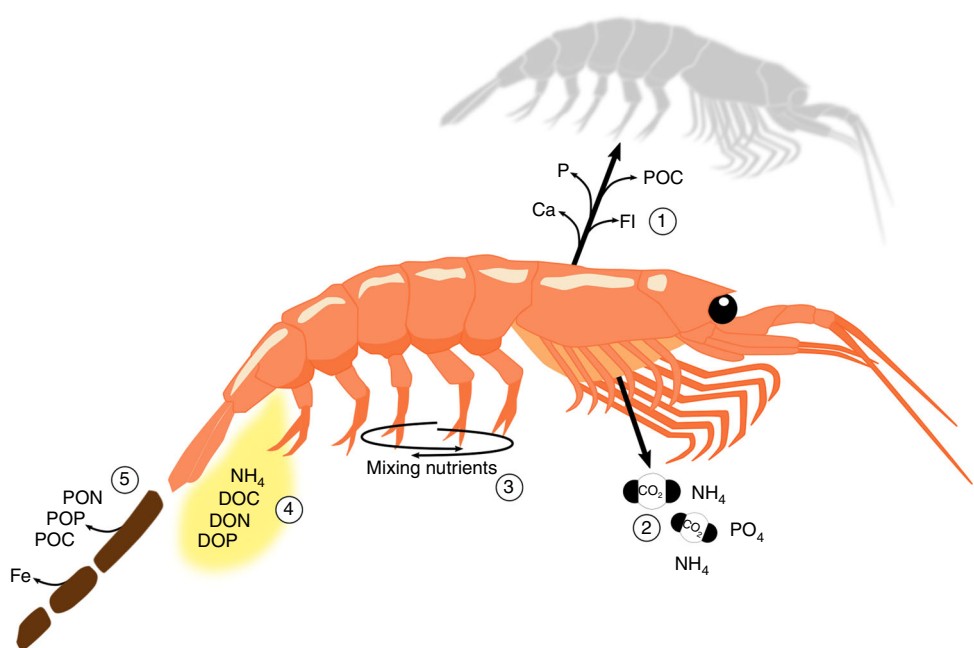

**Fig. 2** Cycling of nutrients by an individual krill. When krill moult they release dissolved calcium, fluoride and phosphorous from the exoskeleton (1). The chitin (organic material) that forms the exoskeleton contributes to organic particle flux sinking to the deep ocean. Krill respire a portion of the energy derived from consuming phytoplankton or other animals as carbon dioxide (2), when swimming from mid/deep waters to the surface in large swarms krill mix water, which potentially brings nutrients to nutrient-poor surface waters (3), ammonium and phosphate is released from the gills and when excreting, along with dissolved organic carbon, nitrogen (e.g., urea) and phosphorous (DOC, DON and DOP, 2 & 4). Krill release fast-sinking faecal pellets containing particulate organic carbon, nitrogen and phosphorous (POC, PON and POP) and iron, the latter of which is bioavailable when leached into surrounding waters along with DOC, DON and DOP (5)

## Krill and biogeochemical cycles

**Carbon**. Pelagic crustaceans such as krill can have a prominent role in regulating the magnitude of carbon stored in the ocean via the biological pump (Fig. 1)[7,25]. During photosynthesis, unicellular phytoplankton transform dissolved inorganic carbon (DIC or $CO_2$), a portion of which originates from the atmosphere, into organic carbon in their cells in the surface ocean[26]

(Fig. 1). Krill feed either directly on phytoplankton, or on protists and invertebrates (mainly zooplankton) that have consumed phytoplankton. A large part of the ingested carbon is absorbed (estimates range from 42 to 94%, dependent on food type and availability[27]), with the remainder being egested via their faecal pellets (Fig. 2). The absorbed carbon components are either catabolised to supply energy (leading to the respiration of $CO_2$),

excreted as dissolved organic carbon[28] or incorporated into body tissue and potentially transferred to krill predators.

Faecal pellets are an integral part of the biological pump[8] (Fig. 1), with some being dense, compact particles that can sink quickly through the ocean. As krill are some of the largest pelagic crustaceans, they produce large faecal pellets (typically up to 1 cm length strings) with variable but often rapid sinking rates[27,29]. Krill pellets constitute the majority of sinking particles analysed in shallow (170 m) and deep (1500 m) Southern Ocean sediment traps deployed west of the Antarctic Peninsula and downstream of South Georgia respectively[30,31]. As krill mostly swarm in vast numbers, their contribution to particulate organic carbon flux can be huge, and estimates span over orders of magnitudes from 7 to 1300 mg C m$^{-2}$ d$^{-1}$[32–35]. However, most observed rates tend to be at the lower end of that range, such as those reported in the marginal ice zone (e.g., 7–104 mg C m$^{-2}$ d$^{-1}$ at 100 m[35]). For reference, values of total (all particle types) particulate organic carbon flux in the Southern Ocean at 100 m, as determined by Thorium-234, ranges from 10 to 600 mg C m$^{-2}$ d$^{-1}$, with an average across latitudes between 100 and 150 mg C m$^{-2}$ d$^{-1}$[36]. In the Scotia Sea (Atlantic Southern Ocean) where krill biomass is high[18], total particulate organic flux at 100 m in the summer is up to 90 mg C m$^{-2}$ d$^{-1}$, with highest fluxes in the marginal/seasonal ice zones[37]. In the marginal ice zone over the productive season, the modelled estimate of the total export flux of krill faecal pellets at 100 m is 0.04 Gt C yr$^{-1}$[35] (equivalent to 42 mg C m$^{-2}$ d$^{-1}$ based on the mean area of the marginal ice zone).

The number of faecal pellets observed generally declines with depth owing to scavenging and degradation[5,6,38] (Fig. 3). However, some studies in the seasonal and marginal ice zones of the Southern Ocean indicate that krill faecal pellets can be transferred extremely efficiently, with minimal attenuation with depth, i.e., the amount of krill faecal pellet carbon in the surface is similar to that at depths of 100 s of metres below[6,37,39]. Such low rates of faecal flux attenuation have not been observed in other oceanic regions or for other crustaceans, suggesting that krill play a disproportionately important role in the sinking of carbon to the deep ocean[35]. Low attenuation of krill pellets in ice regions is

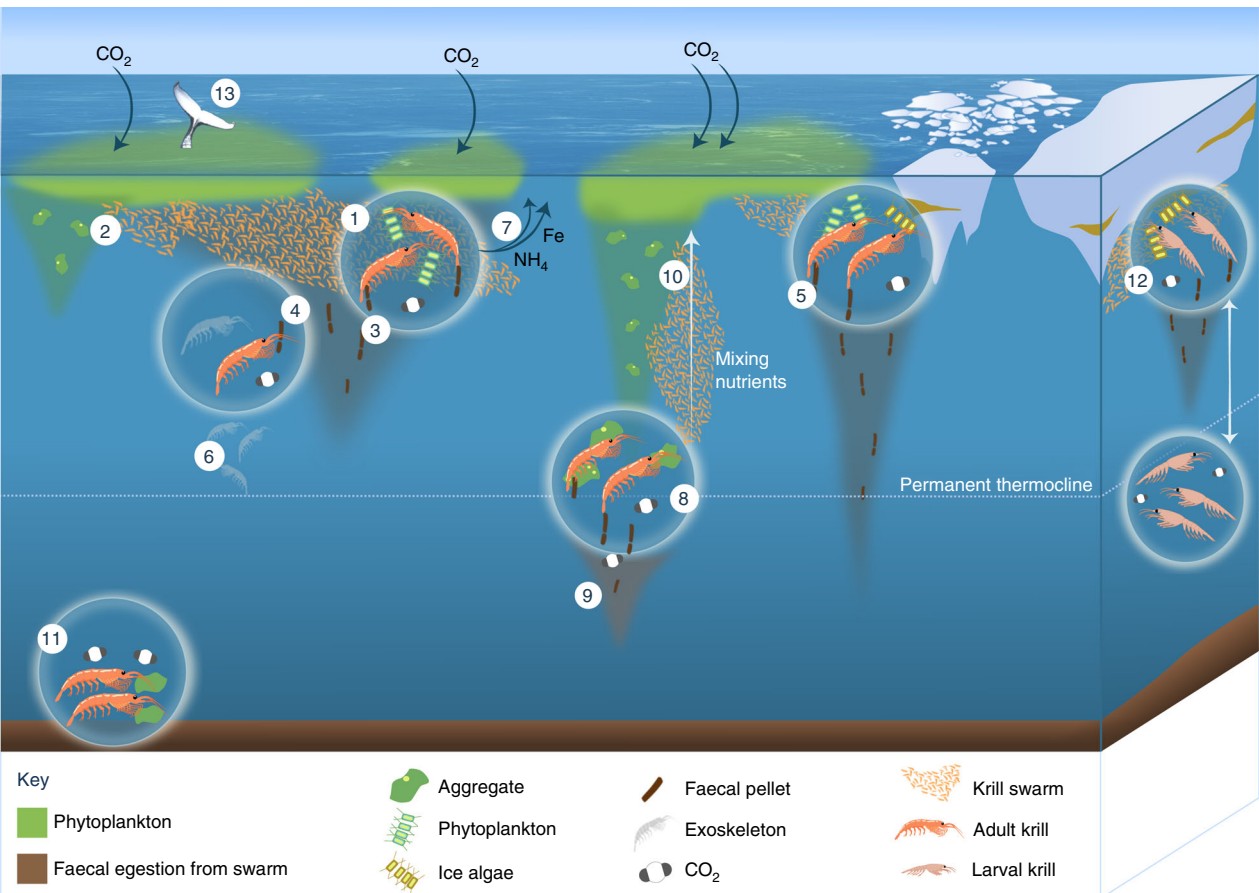

**Fig. 3** Role of *E. superba* in biogeochemical cycles. Krill (as swarms and individuals) feed on phytoplankton at the surface (1) leaving only a proportion to sink as phytodetrital aggregates (2), which are broken up easily and may not sink below the permanent thermocline. Krill also release faecal pellets (3) whilst they feed, which can sink to the deep sea but can be consumed (coprophagy) and degraded as they descend (4) by krill, bacteria and zooplankton. In the marginal ice zone, faecal pellet flux can reach greater depths (5). Krill also release moults, which sink and contribute to the carbon flux (6). Nutrients are released by krill during sloppy feeding, excretion and egestion, such as iron and ammonium (7, see Fig. 2 for other nutrients released), and if they are released near the surface can stimulate phytoplankton production and further atmospheric $CO_2$ drawdown. Some adult krill permanently reside deeper in the water column, consuming organic material at depth (8). Any carbon (as organic matter or as $CO_2$) that sinks below the permanent thermocline is removed from subjection to seasonal mixing and will remain stored in the deep ocean for at least a year (9). The swimming motions of migrating adult krill that migrate can mix nutrient-rich water from the deep (10), further stimulating primary production. Other adult krill forage on the seafloor, releasing respired $CO_2$ at depth and may be consumed by demersal predators (11). Larval krill, which in the Southern Ocean reside under the sea ice, undergo extensive diurnal vertical migration (12), potentially transferring $CO_2$ below the permanent thermocline. Krill are consumed by many predators including baleen whales (13), leading to storage of some of the krill carbon as biomass for decades before the whale dies, sinks to the seafloor and is consumed by deep sea organisms

likely owing to a combination of krill behaviour including pronounced vertical migrations[37,39,40] and the formation of large swarms that produce a 'rain' of fast-sinking faecal pellets that overwhelm detrital consumers[6,27,35,37,39]. In addition, short migrations (40 m) just below the mixed layer can occur multiple times during a night's feed, dependent on the satiation state of the krill[41,42], which may increase the chance of faecal pellet export by shunting pellets deeper into the water column.

Vertical migrations can also shunt carbon to depth when krill occupy deeper layers and respire carbon consumed at the surface, a process termed active carbon flux. This occurs especially in younger developmental stages of E. superba (larvae and juveniles), which can undergo extensive diel (daily) vertical migrations (DVMs)[43,44] travelling to deeper depths than adults, often below permanent thermoclines[41,45] (Fig. 3). Larval DVMs may follow a normal pattern of ascent during the night and descent during the day[46], or a reverse pattern of ascent during the day and descent at night[44]. DVM patterns in adult krill are less clear, and a range of behaviours may be exhibited, including normal and reverse DVM as well as remaining at particular depths throughout the diel cycle[47,48], so their biogeochemical role may differ depending on the depth they inhabit or migrate to. Even so, where DVM does take place in adults, they generally remain above the permanent thermocline, within the surface mixed layer[49]. Difficulties in resolving the complex DVM of Antarctic krill means that estimates of the total contribution of this species to active carbon flux have yet to be fully resolved[41,48,50].

There are further additional mechanisms by which krill might contribute to the carbon sink. For instance, in winter adult E. superba populations appear to move to coastal basins[51] and studies using under-water cameras and active acoustics have revealed that krill aggregate at greater depths in winter than in summer[52,53]. Metabolism of their lipid reserves to $CO_2$ when residing in deeper waters in winter, as observed in copepods[54], releases surface-produced carbon to the deep ocean. This process is termed the lipid pump and is significant in that it moves carbon to depth without depleting surface concentrations of potentially limiting nutrients over winter (e.g., nitrogen and phosphorus). Rapid transport of carbon to the deep ocean/sea floor is also facilitated by the short phytoplankton-krill-whale food chain, where krill carbon is stored as biomass in baleen whales for decades, whose carcasses rapidly sink to the deep sea floor when they die[55] (Fig. 3). Finally, some E. superba also feed on detritus on the seabed, often at great depth, and are then fed upon by benthic fish and invertebrates meaning the carbon stays in the deep ocean[50] (Fig. 3). The contribution of all these processes to carbon transport is potentially significant but remains unquantified.

**Iron**. Iron is an important trace element in the oceans and its low availability limits primary productivity in large areas, including much of the ice-free Southern Ocean[56,57]. The largest sources of new iron to the Southern Ocean surface waters are deep winter mixing[58] and the seasonal melting of sea ice[59]. Following the depletion of this winter-spring iron pulse, further primary production depends increasingly on recycled iron[58]. E. superba have an important role in oceanic iron cycling[49,50,60–62] facilitated by the ingestion of iron-rich phytoplankton and lithogenic particles. The iron concentration in an individual whole adult krill ranges from 4.4 to 190.5 mg kg$^{-1}$[49,50,60,61,63,64], with the > 40-fold difference in krill iron content reflecting seasonal and regional differences in their dietary iron content[64]. Eventually, the iron retained in individual bodies can be released back into surface waters when baleen whales and other vertebrates consume E. superba and subsequently defecate[60]. Thus, in the iron-limited

Southern Ocean iron recycled via krill and their predators is important for stimulating primary production (Fig. 3).

A small proportion of dissolved iron (dFe, < 0.2 μm[49]) in excess of the demand by E. superba is excreted, with excretion rates ranging from 0.2 to 5.5 nmol dFe ind$^{-1}$ d$^{-1}$[49]. Highest rates occur when krill feed on diatoms, which is consistent with some diatoms' ability to acquire and store excess intracellular iron[65]. Upon digestion of phytoplankton, E. superba may also release iron-binding ligands (e.g., porphyrin compounds)[9], which can complex with inorganic iron and thereby increase the concentration of soluble iron available to phytoplankton[49]. However, most (90%) of the iron in E. superba is released via their fast-sinking faecal pellets, which have 3–4 orders of magnitude more iron their muscle tissue (Fig. 2)[49]. Therefore, the cycling of iron via krill is closely linked to the fate of their faecal pellets, which may sink to great depths without being consumed[37,39]. A study on salps showed that iron was not readily leached from their faecal pellets[66], and, if also true for krill, their pellets would need to be fragmented to release dFe into the water column as the pellet sinks. Nevertheless, the feeding activity of the abundant E. superba as a whole provides the basis for several pathways of dFe supply to phytoplankton (Fig. 3)—involving also microbes, zooplankton and krill predators—which, together with the release of ligands, can benefit phytoplankton growth. Such fertilising processes mediated by krill may explain why phytoplankton blooms downstream of the island of South Georgia last longer and are more intensive during years with high krill abundances on-shelf[49].

**Macronutrient regeneration and grazing**. Krill also release macronutrients such as ammonium (Fig. 3), which can be particularly important in iron-limited regions, as using ammonium rather than nitrate reduces the phytoplankton iron demand by ~ 30 %[67]. Regions of frequently high but spatially variable E. superba density have been used as a series of natural experiments in examining the role of krill nutrient recycling and grazing in shaping the abundance and composition of phytoplankton. At South Georgia, grazing was sufficient to suppress phytoplankton biomass toward the east of the island[68], yet E. superba ammonium excretion also supplied a large fraction of the requirements to the ungrazed cells. Rates of ammonia excretion in South Georgia have been measured to range from 12 to 273 nmol NH$_4$ ind$^{-1}$ h$^{-1}$[69], with higher rates measured further south off the Western Antarctic Peninsula (61–475 nmol NH$_4$ ind$^{-1}$ h$^{-1}$)[70].

In addition, E. superba grazing and deep mixing has been found to shift the phytoplankton community from diatoms to flagellates at the Antarctic Peninsula[71]. Krill grazing can also fragment phytoplankton cells or other particulate matter releasing dissolved organic matter into the water (termed sloppy feeding, Figs. 1 & 2)[38,72], which can be further broken down and remineralised by bacteria (termed microbial gardening[73]). This process reduces the flux of carbon to the deep ocean, although, thus far a link between sloppy feeding and increased microbial activity has not been explicitly shown for krill. E. superba can thus exert two opposing top–down controls on phytoplankton; they can rapidly graze blooms decreasing phytoplankton biomass but also excrete nutrients increasing phytoplankton biomass.

**Transport of nutrients**. In addition to shunting carbon to deeper waters, krill are also involved in the vertical and lateral transport of other nutrients. For instance, adult E. superba moult as often as every 2 weeks depending on temperature and season[74], resulting in a high number of moults produced per krill over their long life-span (5–6 years in wild). The release of moults, which sink at rates of 50–1000 m d$^{-1}$[75], contributes to the carbon sink, but also

to the release of other micronutrients to the water column as the moult sinks. For instance, fluoride concentrations in live *E. superba* exoskeletons are at least 2500 times higher than the surrounding waters[76], and this fluoride is leached out during ecdysis[75] and degradation of the exoskeletons. A range of other elements are also found in the exoskeletons of krill, for example the exoskeleton contains 47% of the phosphorous and 84% of the calcium concentrations of these minerals in krill[77]. How quickly these nutrients are released from shedded exoskeletons (moults) and their possible contribution to biogeochemical cycles has yet to be quantified.

Krill can also mix nutrients; mass migrations of krill swarms from deep nutrient-rich water, particularly in localised, permanently or temporarily oligotrophic waters, could mix nutrients to the surface and stimulate phytoplankton growth[42,78] (Fig. 3). Conversely, the carbon transferred by krill from the surface to below the mixed layer is subjected to remineralisation by bacteria and detritivores, which convert dissolved organic carbon to $CO_2$[6,38]. The depth at which this remineralisation occurs, or the depth of krill respiration, is crucial for determining the longevity of $CO_2$ storage in the deep ocean; i.e., whether the released $CO_2$ is mixed back up to the surface (shallow remineralisation) or is stored for decades in the deep ocean (deep remineralisation)[79]. If $CO_2$ is released above the permanent thermocline (deepest winter mixed layer depth, globally <750 m[80]), then $CO_2$ will be subjected to seasonal physical mixing to the surface ocean and potentially re-exchanged with the atmosphere within a year following release from the krill (Fig. 3). The length of time $CO_2$ (or nutrients) will remain in the deep ocean also depends on the water mass it enters owing to ocean circulation[81]. For *E. superba* that live south of the Antarctic Circumpolar Current (ACC, i.e., a substantial part of the population[82]), any nutrients they release will likely remain in the Southern Ocean. However, nutrients released from an organism within the ACC, or at the northern boundary of the ACC, may be subducted into the Antarctic Intermediate Water. Currently we do not know whether nutrients released by Southern Ocean organisms make a significant contribution to production elsewhere.

**Larval stages**. The contribution of larval krill to biogeochemical cycles is different to that of adults due to their unique pattern of growth and development, smaller size and feeding ecology[83]. Larval *E. superba* use sea ice as a feeding ground and shelter[84] and owing to their ingestion of ice biota and subsequent migration into the water column, play an important role in ice-pelagic coupling. *E. superba* larvae consume up to 26% of their body weight in carbon per day, of which ~ 10% is egested as faecal pellets[85]. This equates to larval egestion of ~ 4 μg C d⁻¹, which is ~ 1000 times less than adults[41] although in the Scotia Sea they can be up to 100 times more abundant than adults[86]. If these relative abundances hold across the wider ocean sector, this would equate to larvae contributing an additional 1–10% of the adult faecal pellet flux. Furthermore, DVM in larval *E. superba* takes them considerably deeper than adults (400 m and 200 m, respectively)[43,48]. The pronounced DVM patterns of larval krill in the proximity of ice may be responsible for the low attenuation of krill faecal pellets with depth in the marginal ice zone of the Atlantic Southern Ocean[6,37] (Fig. 3), rather than the DVMs of adult krill. Larvae may be more likely to contribute to active transport of carbon via egestion and respiration at depth, although the mass and sinking potential of larval faecal pellets have yet to be characterised.

In summary, *E. superba* influence many biogeochemical cycles including carbon, nitrogen and iron, from larval through to adult life stages, and also have a diverse, multi-faceted role within these individual elemental cycles. Whilst there has been some focus on

the contribution of *E. superba* to organic carbon and iron cycles, given our current lack of knowledge and uncertainty in biomass estimates, (Box 1) it is difficult to quantify its complete role in these cycles. Nonetheless, the substantial biomass, diurnal vertical migrations and broad horizontal distribution of *E. superba* suggests a significant contribution. Quantification of these rates, as well as better constrained estimates of krill biomass, are critical to provide meaningful data so biogeochemical modellers can sufficiently parameterise the influence of *E. superba* on nutrient cycles. A better understanding of krill–nutrient interactions will also allow assessment of the impact of human activities, particularly fishing, on biogeochemical cycles and help to identify management approaches that will minimise these impacts.

## Implications of declining *E. superba* biomass

The complex biogeochemical roles of *E. superba* means that harvesting krill (Box 2) could have variable and potentially opposing effects on ocean biogeochemistry. In this section, we detail the possible impacts harvesting krill could have on the Southern Ocean carbon sink given current knowledge. We also briefly discuss the biogeochemical implications of potential changes in krill biomass owing to the recovery of whale populations and to climate change.

As discussed, large, fast-sinking krill faecal pellets can form a large proportion of total particulate organic carbon flux in the Southern Ocean[6,37]. If krill are removed from the ecosystem, this faecal pellet flux will decrease. To estimate the reduction in this sink owing to the removal of *E. superba* by the Atlantic Southern Ocean fishery (Box 2), we assume pellet production and attenuation rates as reported in Belcher et al.[35] (3.2 mg C ind⁻¹ d⁻¹ and 0.32, respectively) and use a mean (2014–2018) annual *E. superba* catch of 264,505 tonnes in Area 48 (Fig. 4). We estimate that the decline in the *E. superba* faecal pellet carbon flux at 100 m over spring and summer due to fishing is 0.6–0.8 mg C m⁻² d⁻¹, with the range incorporating pellet egestion at 40 or 80 m, respectively. Total pellet fluxes from krill and other zooplankton can be up to 78 mg C m⁻² d⁻¹ in the Southern Ocean marginal ice zone[37], thus this average decline in pellet flux is fairly low because only <0.5 % of the *E. superba* population is caught by the fishery. If the trigger- or catch limits in Area 48 (0.62 Mt yr⁻¹ and 5.61 Mt yr⁻¹, respectively) were met, then fishing would result in a respective decline of the *E. superba* pellet flux by 1.5–1.8 mg C m⁻² d⁻¹ or 13.1–16.7 mg C m⁻² d⁻¹. These calculations contain certain assumptions (e.g., krill egestion depth, pellet egestion and attenuation rate), but are presented as a thought experiment to highlight how the magnitude of the pellet carbon flux may change if the fishery expanded.

The above faecal pellet flux calculations consider adults only and do not include any active transfer via DVM or the lipid pump. As larval *E. superba* migrate deeper in the water column than adults they may egest pellets deeper, resulting in a lower attenuation rate (< 0.32) of larval pellet flux and a higher proportion of carbon reaching the deep sea, even though their pellets are likely smaller and potentially sink slower than those of adults. Given that migrating larvae will also respire $CO_2$ deeper and eventually become the next generation of adults, larval biomass is important to protect. Currently, the fishery only targets adult *E. superba*, but we recommend measures are put in place to ensure that as fishing technology advances, the fishery does not further encroach on larval habitat (i.e., near the sea ice) and precautions are taken to prevent larval bycatch when nets become clogged by adult *E. superba*. As it is not currently known if larvae are caught by the fishery, CCAMLR should initiate research to determine whether the current fishery is catching larval krill as bycatch and, if so, adopt regulations to prevent this happening in the future.

### Box 2. | The *E. superba* fishery

*E. superba* have been fished in the Southern Ocean since the 1970s, to produce meal used as aquaculture feed and oil for human consumption. Catches peaked at 530,000 tonnes (t) yr$^{-1}$ in the 1980s and declined with the collapse of the Soviet fishing industry, but have since increased steadily to over 306,000 t yr$^{-1}$ in 2018 (Fig. Box 2)[136]. The fishery is managed by CCAMLR (Commission for the Conservation of Antarctic Marine Living Resources), and a range of Conservation Measures regulate mandatory notification of intention to fish, minimisation of seal and bird bycatch, reporting, scientific observation and annual catch limits[19]. CCAMLR is mandated to apply ecosystem-based management[137] with no explicit measures to regulate fishery impacts on biogeochemical cycles. Rather, management relies on catch limits that are low relative to estimates of pre-exploitation biomass.

**Precautionary catch limits on the krill fishery[136]**

| Conservation measure | Area/Division | Total catch limit (Mt) | Trigger level (Mt) | Pre-exploitation biomass (Mt), (year of estimate) |
|---|---|---|---|---|
| 51-01 | 48.1, 48.2, 48.3, 48.4 (Southwest Atlantic) | 5.61 | 0.62 | 60.3, (2000) |
| 51-02 | 58.4.1 (Eastern Indian) | 0.44 | n/a | 4.8, (1996) |
| 52-03 | 58.4.2 (Western Indian) | 2.645 | 0.452 | 27.8, (2006) |

*Mt* million tones

The catch limits total < 10 % of estimated adult *E. superba* biomass in areas open to fishing. CCAMLR has also established much lower trigger levels in most surveyed areas, that cannot be exceeded until sufficient information is available to avoid localised concentration of the catch. In the southwest Atlantic this trigger level is 620,000 t yr$^{-1}$ (~1 % of the estimated pre-exploitation biomass) and can only be changed by a consensus decision by all Members of the Commission. The trigger level in the Southwest Atlantic sector, where almost all current fishing occurs[19], has been subdivided into further catch limits in each of the individual subareas and for the last 4 years the catch in subarea 48.1 (Fig. 4) has reached the 155,000 tonne limit and the subarea has been closed for the rest of the season[138].

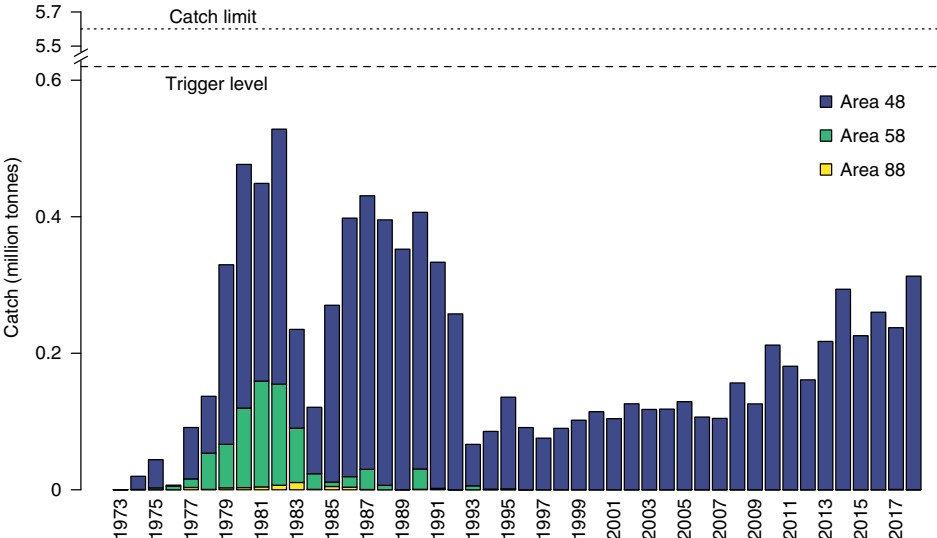

**Fig. Box 2** Fishery catch (million tonnes) of *Euphausia superba* in the Southern Ocean from 1973 to 2018, from three CCAMLR areas — 48, 58 and 88. Management trigger level (0.62 million tonnes) and catch limit (5.6 million tonnes) also shown. Data from CCAMLR[136].

Fishery-driven declines in adult *E. superba* biomass could result in opposing effects on the carbon sink. As *E. superba* consume phytoplankton and the subsequent phytodetrital flux, a decline in *E. superba* could reduce grazing, potentially increasing the proportion of phytoplankton biomass that is exported as phytodetrital flux (process #2 in Fig. 3). The ratio of phytodetrital aggregates to faecal pellets in particle flux varies globally but is generally low in the Southern Ocean where pellets dominate[6]. However, we do not know to what extent removing krill may increase the magnitude of aggregate flux. Whilst removing krill may increase phytoplankton biomass through a decline in consumption rates, it would also decrease the fertilisation effects of krill (e.g., ammonium excretion) that have been observed to increase phytoplankton biomass. As we are unable to quantify many of the other aspects of the carbon cycle (e.g., contribution of *E. superba* to active transfer of carbon, or an increase in

aggregate flux), it is not clear what the exact effect of removing *E. superba* via the fishery would be on the carbon sink, and this is a key question that needs to be answered.

Another unknown regarding the impact of the fishery is which zooplankton (i.e., copepods, salps etc.) could replace the ecological niche left by any decline in *E. superba*. Copepods, which have a higher production rate than *E. superba*[46,87], also recycle iron[88], produce sinking faecal pellets and contribute to the Southern Ocean lipid pump[87]. Yet, *E. superba* have a different biogeochemical role than the dominant (smaller) grazers in the Southern Ocean. The size of krill gives them a strong swimming ability allowing swarm formation, which we speculate contributes to their efficiency in pellet export, alongside large pellet size and associated high sinking speed. Also relating to body size is an ability to feed on a very wide range of particle sizes using a large but fine mesh filter[46], including large, iron-storing diatoms and

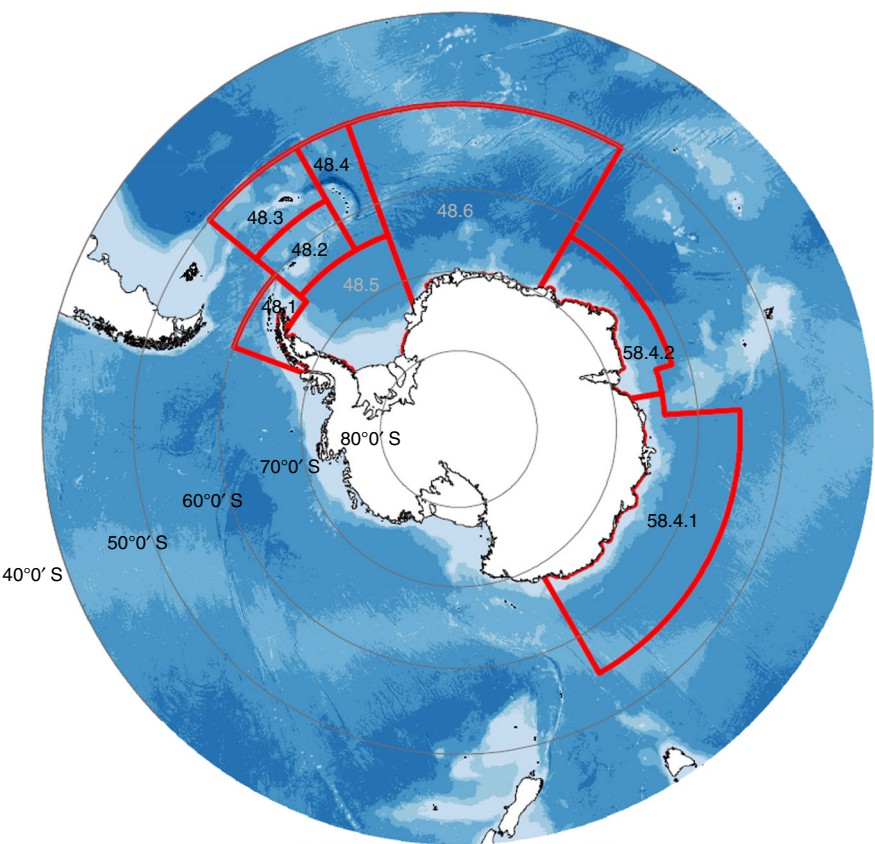

**Fig. 4** Map of Antarctica and fishing areas for *E. superba*. Commission for the Conservation of Antarctic Marine Living Resources (CCAMLR) Area 48 is subdivided into six smaller areas (48.1–48.6) covering the Atlantic sector of the Southern Ocean. Numbers in grey are subareas not fished and those in black are fished. Bathymetry is shown in blue with dark blue, representing deeper waters (data from the GEBCO_2014 Grid, version 20150318, http://www.gebco.net). The Antarctic coastline was obtained from the Scientific Committee on Antarctic Research (SCAR) Antarctic Digital Database

lithogenic particles. While these functions of recycling nutrients are partially available to smaller grazers[49], the size and swarming behaviour of krill results in different impacts on biogeochemical cycles than zooplankton. Thus, the biogeochemical role of krill would not be replaced like-for-like by copepods. Salps are non-selective feeders, can exist in swarms and have fast-sinking pellets and caracases[89] and thus could potentially fill part of the biogeochemical niche of krill, if they replaced *E. superba* biomass removed by the fishery. Although salp particle flux is generally thought to be high, it can be variable with high attenuation rates observed in the Southern Ocean[90], and thus the contribution of salps to biogeochemical cycles is an area for future research.

Krill biomass is also influenced by the abundance of their predators. Exploitation of Southern Ocean organisms since the late 1700s has severely perturbed the krill foodweb and thus krill biomass, via the sequential extraction of fur seals, baleen whales and endemic fish species[91,92]. For example, the reduction in whales potentially increased krill grazing pressure on phytoplankton (diatoms in particular[92]), and so decreased iron recycling[93]. With the rapid recovery of baleen whales[94] *E. superba* biomass may further decline, although it has been suggested that a large whale population feeding on krill might increase the recycling of iron in surface waters thus increasing Southern Ocean productivity[95]. Because ecosystem change associated with recovering baleen whales is occurring alongside human-driven warming it will be complicated to tease apart the factors that might be changing krill biomass in the future. There is an urgent need to better understand the recovery of baleen whales in the Southern Ocean and the ecological consequences of their return.

There are concerns for the future of *E. superba* in a rapidly changing climate. The Southwest Atlantic sector warmed rapidly during the last century[96,97], and this is both the main population centre for *E. superba* and is where the Antarctic fishery is concentrated. There are reports of declines in krill density within this sector[98–101], particularly in the northern part of the Southwest Atlantic, with evidence of a more stable population toward the south, including over the continental shelf of the Western Antarctic Peninsula[101]. Recruitment to juvenile *E. superba* has declined rapidly over the last 40 years, associated with increasing positive anomalies of the Southern Annular Mode[101]. Probably coupled with changes in mortality, this has resulted in a 75% increase in mean body mass of the post-larval population[101]. Population dynamic models predict further declines in krill populations, particularly around the high phytoplankton biomass and carbon export region of South Georgia[97,102]. With the possible exception of increased acidification[103], most of the projected future climate change scenarios, such as trends in temperature, sea ice cover and climatic modes are likely to have a negative impact on adult *E. superba* biomass[104–106].

Each of these observed and potential future changes has implications for biogeochemical cycles. A larger mean body size of the krill population may increase pellet size and sinking speeds, yet a decline in krill biomass is likely to reduce the role of krill in biogeochemical cycling. Investigating how perturbing the foodweb via commercial harvesting, together with climate change and changing predator populations, is important to assess the future state of biogeochemical cycles.

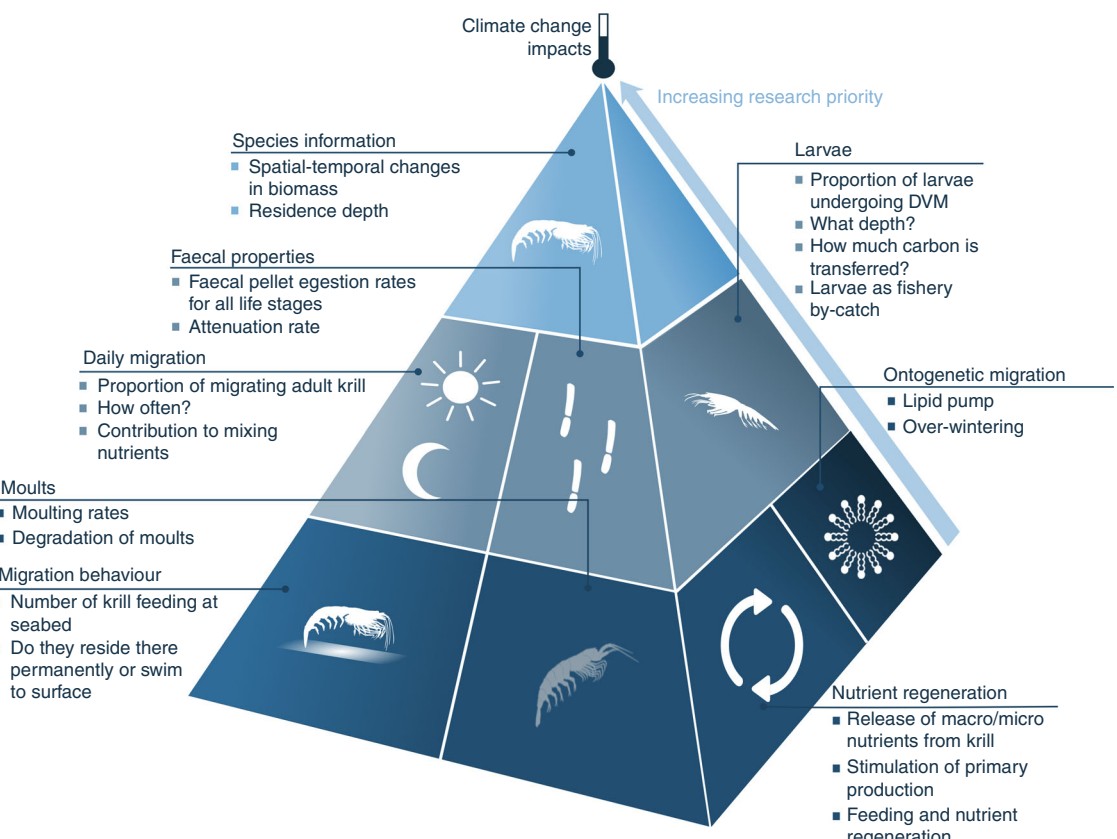

**Fig. 5** Future research priorities on *E. superba* processes for the next decade. The pyramid illustrates the rates and states of krill life history, habitat and biogeochemical function, which need to be prioritised as areas of key research in the coming decade. The underlying ones, namely biomass (including spatial and temporal variations) and the residence depth of krill, are listed at the top of the pyramid. Determining the influence of climate change on all of these processes is vital. Autonomous vehicles fitted with biogeochemical sensors and cameras and acoustics will be instrumental in collecting data on this cryptic species. These key areas of research are needed to be able to parameterise krill in both ecosystem and biogeochemical models

## Future directions

Compiling our current knowledge of *E. superba* in biogeochemical cycles has highlighted some key areas of future research in order to quantify fully the role of *E. superba* in biogeochemical cycles. This includes some hitherto neglected areas, such as the contribution of larval *E. superba* to the Southern Ocean carbon sink (Fig. 5) and determining whether larvae are currently part of the *E. superba* fishery bycatch, as well as including some areas where progress has started to be made. These include quantifying the extent to which adult *E. superba* may enhance primary production through fertilisation and identifying the other species that may replace the ecological and biogeochemical niche left by *E. superba*. In addition, determining if krill contribute to mesoscale nutrient mixing requires combining laboratory mixing experiments with estimates of vertical migrations, and answering such questions as; what proportion and what age (e.g., adults vs. larvae) of the population migrates, what depth do they migrate from, do they migrate every day and/or seasonally, and, if feeding at the seabed, do they reside there permanently or swim to the surface too?

Whilst there is a clear need for more understanding on the role of *E. superba* in biogeochemical cycles, of equal importance are more accurate estimates of their biomass (Fig. 5) and distribution in the water column. As discussed in Box 1, sampling for krill is sporadic, seasonal, spatially and temporally patchy and based on different methods (nets or acoustics), contributing to uncertainty in krill biomass estimates. Other key information surrounding biomass include the biomass of larval krill (in open water and

under ice), the daytime residence depth of all krill life stages and the proportion of krill undergoing ontogenetic migrations. Better estimates of biomass would help constrain the potential scale of the impact krill have on all biogeochemical cycles. These could be achieved through improved conversion factors between acoustic return and biomass, experimental comparisons between the results of nets and acoustics, synoptic surveys in regions outside of the Atlantic sector of the Southern Ocean, and new technologies such as remote acoustic samplers and long-range sonars[107] and cameras attached to remotely operated vehicles[52]. These should be combined with biogeochemical experiments to determine the biogeochemical function of different krill populations residing at or transiting to different depths. To produce circumpolar *E. superba* biomass estimates, data assimilation and syntheses may be necessary. Some key parameters needed for this are food availability (e.g., satellite primary production), predator consumption rates, population estimates, suitable habitat volume and maximum swarm density. These approaches have been attempted individually with varying success[108–110], but are yet to be assimilated and used together to estimate *E. superba* biomass. Primary production can provide an upper limit on krill population biomass[110], but foodweb models are needed to fill gaps where data are sparse and variable for krill predators[111].

Foodweb and fishery models are also useful for investigating how krill biomass may change owing to fishing pressure, climate change or predator biomass changes[112,113]. Lower trophic levels (including krill) are particularly under-represented in Southern Ocean foodweb models, with krill, mesozooplankton and trophic

| Reference | Model base | Modelled system | FGs | P FGs | Z FGs | Aa krill own FG? | Aa krill life stage FG? | Aa krill predators | % lower trophic level FGs |
|---|---|---|---|---|---|---|---|---|---|
| 115 | Ecopath | Prydz Bay | 28 | 1 | 5 | Yes | No | 15 | 28% |
| 119 | Ecopath | Antarctic Peninsula | 39 | 1 | 1 | Yes | Yes 2 life stages | 17 | 10% |
| 120 | Ecopath | Antarctic Peninsula | 28 | 1 | 2 | Yes | Yes 2 life stages | 17 | 18% |
| 121 | Ecopath | Antarctic Peninsula | 58 | 4 | 4 | Yes | Yes 4 life stages | 31 | 21% |
| 122 | Ecopath | Antarctic Peninsula | 24 | 3 | 6 | Yes | Yes 2 life stages | 10 | 46% |
| 123 | Ecopath | Antarctic Peninsula | 63 | 4 | 8 | Yes | Yes 4 life stages | 30 | 25% |
| 124 | Ecopath | South Georgia | 30 | 1 | 4 | Yes | No | 10 | 20% |
| 125 | Ecopath | South Georgia | 30 | 3 | 4 | Yes | No | 17 | 27% |
| 126 | Ecopath | The Ross Sea | 38 | 3 | 6 | Yes | No | 13 | 26% |
| 127 | Ecopath | Southern Ocean | 18 | 1 | 4 | Yes | No | 11 | 33% |

**Table 1 Summary of lower trophic level representation (krill, zooplankton and below) in Southern Ocean foodweb models**

Krill are not included in the zooplankton FGs. *FGs* functional groups, *P* phytoplankton, *Z* zooplankton, *Aa* Antarctic

levels below representing a small proportion (mean = 25%, Table 1) of the total functional groups. Aggregation of lower trophic levels is often necessary to reduce uncertainty related to incomplete trophic information[112]. These models (Table 1) are focused on the Antarctic Peninsula where the information required to parameterise krill life stages is available, while system-specific information for krill is unavailable for regions such as the Indian sector where few foodweb models exist[114]. Whilst *E. superba* are often parameterised as a separate functional group in Southern Ocean foodweb models (Table 1), the same cannot be said for biogeochemical models, where typically krill are incorporated into a large zooplankton pool[115]. The benefit of biogeochemical models though is that the low trophic level organisms (e.g., phytoplankton and krill) interact with nutrients in the surrounding water, whereas foodweb models only capture trophic interactions. With temperature and nutrient concentrations projected to change globally[116], more progress is necessary on coupling foodweb and biogeochemical models to build an end-to-end model from nutrients to top-predators[117]. A current example for the Southern tracks organic carbon through the ecosystem from phytoplankton to penguins, including an explicitly parameterised krill group[118]. Ideally end-to-end models will be able to incorporate nutrients and physical water circulation to make spatial projections of the impacts of climate change and fishing on biogeochemical cycles and vice versa.

To understand fully the role of a fishery or the changing climate on *E. superba* and biogeochemical cycles we need information on krill in today's environment but also on how they will behave and fare in a warmer more acidic ocean. We also need to account for spatial heterogeneity, because climate change may make lower latitudes uninhabitable and open new habitats in the south. Laboratory experiments should be combined with newer and advancing technology such as autonomous vehicles (e.g., ocean gliders or autosubs) and camera systems[52], which could simultaneously measure critical nutrients such as nitrate, iron etc. and image krill swarms and behaviour in situ. In addition, linking krill behaviour and biomass/location to features that can be measured by satellite (e.g., temperature, ocean colour and sea ice) would provide large spatial and temporal coverage that could be used in models. Using multiple approaches to study krill will be vital to gain all the information needed on these cryptic organisms. Collaborating with the krill fishing fleet by providing them

with biogeochemical sensors, and possibly autonomous vehicles, would widen the temporal and spatial coverage of data (Box 2).

## Summary

The large body size, high biomass and swarming ability of *E. superba*, coupled with physiological traits such as large faecal pellets and excretion into nutrient-limited waters, means *E. superba* has a prominent role in the cycling of nutrients in the Southern Ocean. The vertical migratory habits of *E. superba* throughout the water column, a trait particularly prominent in larvae but more complex in adults, shows they can influence both the deep carbon sink and stimulate surface primary production. As the Southern Ocean has a disproportionately important role in the global carbon sink, and productivity is limited in iron-deplete areas, the cycling of carbon, iron and ammonium by *E. superba* has a particularly significant role compared with krill in other regions. However, the life-history traits of all krill (e.g., large body size, swarming ability) potentially means other krill species are important in biogeochemical cycles.

We have shown that the role of *E. superba* in biogeochemical cycles is significant, but uncertain: this uncertainty extends to the times and locations when biogeochemical activity is most intense and to the magnitude of the role of *E. superba*. Particularly crucial are ongoing efforts to estimate the absolute *E. superba* biomass and determine their residence depths and migration patterns, including that of larvae.

Our lack of knowledge of the true extent of krill's ability to regulate biogeochemical cycles is a concern given *E. superba* are the target of the largest fishery in the Southern Ocean. Whilst the *E. superba* fishery is managed and regulated by CCAMLR (Commission for the Conservation of Antarctic Marine Living Resources), there has been no active consideration of the biogeochemical role of krill by CCAMLR or to our knowledge the biogeochemical effect of any other managed fishery. Globally, measures to maintain biomass and productivity of stocks of fished species indirectly help to preserve their biogeochemical role. However, fishery management needs to consider the influence of harvesting on biogeochemical cycles. *E. superba* biomass and their biogeochemical role are both likely to be impacted by the activity of fisheries and climate change, with uncertain implications for future biogeochemical cycles.

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

## Acknowledgements

This publication was funded by a Pew Charitable Trust grant awarded to E.L.C., and E.L.C. was funded by an Australian Research Council Laureate (FL160100131) awarded to P.W.B. A.B., G.A.T. and S.L.H. were supported by the Ocean Ecosystems programme at British Antarctic Survey, and A.B. and G.A.T. additionally by a Large Grant from the UK Natural Environment Research Council (NERC, NE/M020835/1). D.K.S. was supported by the US National Science Foundation's Antarctic Organisms and Ecosystems Program (grant PLR 1440435). Contributions from A.A. and K.S. were funded by the UK NERC through PICCOLO (RoSES programme) NE/P021409/1. The graphics (Figs. 1–3 and 5) were produced by McCork Studios.

## Author contributions

E.L.C. conceived the manuscript. E.L.C. and A.B. lead the writing and figure design. D.K.S., L.R., K.S., B.M., S.K., S.M. and P.W.B. contributed to the writing of the manuscript, with significant contributions from G.A.T., A.A., S.N. and S.L.H.

## Competing interests

Steve Nicol has been employed to provide scientific advice to the Association of Responsible Krill harvesting companies.

## Additional information

**Peer review information** *Nature Communication* thanks Cassandra Brooks and other, anonymous, reviewers for their contributions to peer review of this work. Peer review reports are available.

