## [Peer Review File · Nature Communications]

Reviewers' comments:

Reviewer #1 (Remarks to the Author):

Cavan et al. present a review of the role of *Euphausia superba* in Southern ocean biogeochemistry. I agree with their central argument that not enough is known about the biogeochemical impact of *E. superba* (although it is important to note that this is not simply due to a lack of research, but also stems from conflicting results about, e.g., whether and/or when *E. superba* engages in DVM behavior). The authors give a thorough treatment of relevant studies of *E. superba*. However, they show a remarkable lack of nuance in discussing and interpreting the results of these studies. This lack of nuance runs through most of the manuscript, but is starkest in their "case study." The authors' discussion in this section basically can be summarized as: krill biomass in the region is X . Each krill produces Y amount of fecal pellet carbon per day and Z percent of these pellets sink past a depth of 100 m (or 1000 m). The flux of carbon from krill pellets is thus $F = X*Y*Z$. If 1% of these krill are removed by the fishery, carbon flux in the Southern Ocean is decreased by 1% times F . By comparing to the traded price of carbon (lines 401 – 402) it is clear that the authors are implying that catching 1% of krill biomass leads to a decrease in carbon export in the Southern Ocean by 1%* F . This ignores everything we know about how ecosystems behave and obfuscates the true issues that need to be investigated with respect to *E. superba* biogeochemical impact. Specifically, 1) the relatively short lifetimes of *E. superba* in the wild, suggest that 1% / year catch limits will not be likely to substantially change the biogeochemical impact of *E. superba* (in part because the biogeochemical impact may be limited by bottom-up food availability rather than *E. superba* abundance) and furthermore the interannual variability in *E. superba* distributions driven by natural climate cycles and/or anthropogenic forcing is certain to have a greater impact than fishing. 2) Removal of *E. superba* by fisheries does not mean that the carbon that would have been exported by fecal pellets will instead just go right back into the atmosphere. *E. superba* fecal pellet production is supported by grazing on diatoms and other organisms. If *E. superba* densities are decreased, the phytoplankton production will be shunted elsewhere in the ecosystem (and total phytoplankton production may even increase if the ecosystem is wasp-waisted as the authors suggest). Other potential fates of this phytoplankton production include (but are certainly not limited to): aggregation and direct sinking of phytoplankton (potentially greater carbon export efficiency, since no carbon is lost to respiration of krill), consumption by salps or other mesozooplankton (likely similar carbon export efficiency), or consumption by protistan zooplankton (likely reduced carbon export efficiency). I elaborate on why the authors must consider ecosystem responses in detailed comments below.

Throughout this review, the authors seem to be starting with the assumption that *E. superba* plays an important role in increasing the magnitude of the biological pump and hence consistently point out the magnitude of fluxes mediated by *E. superba*. These fluxes are indeed substantial (because the biomass of *E. superba* is massive). However, the authors ignore evidence that perhaps *E. superba* actually suppresses the biological pump. Such evidence needs to be discussed. First, from a conceptual framework, it is often suggested that zooplankton grazing decreases the magnitude of the biological pump because without grazing potentially 100% of NPP can be exported, while grazers will respire a large proportion of what they consume leading to decreased export efficiency (e.g. Buesseler & Boyd, 2009). Furthermore, while in situ sinking rate measurements do indeed find high settling velocities when krill pellets dominate flux, they can often show equally high velocities when larger aggregates dominate the flux (McDonnell & Buesseler, 2010). Furthermore, in studies that have looked at krill abundance and carbon export, there is no positive correlation between export efficiency and krill abundance (indeed, limited evidence suggests a negative correlation may be more likely, see e.g., Fig. 7B of Ducklow et al. 2006 or Fig. 19 of Stephanie Owens' MIT dissertation). Furthermore, as the authors have noted, *E. superba* abundance may actually depress total primary production, which will likely decrease the magnitude of the carbon pump.

The authors devote one third of the abstract to stating that the Antarctic krill fishery must consider biogeochemical impacts of removing krill. In light of the (relatively minor) impact found by the

authors' own calculations and the comments I have made above, I do not think that this sentence is justified.

Line 155-157: The authors state that salp fecal pellet flux attenuation in the SO is typically higher than that of krill. The studies they cite show: 1) a highly variable flux attenuation for krill pellets across many studies and 2) relatively low salp pellet attenuation in a single study. When you combine 1) the high variability in krill pellet attenuation, 2) the paucity of studies of salp pellet attenuation in the SO, 3) the general expectation that the rapidly sinking salp pellets will lead to low pellet attenuation, and 4) the repeated finding in other ecosystems that salps lead to high carbon flux to the benthos (e.g. Pfannkuche & Lochte, 1993; Smith et al. 2014), the statement that salp attenuation is "typically" higher than that of krill is unjustified. Quite simply, we don't have enough data to know what is typical.

The authors seem to imply that *E. superba* DVM may be a substantial flux of carbon to depth. However, the results of multiple studies leave a lot of uncertainty as to whether there is a consistent diel pattern in the species. Furthermore, the studies that have documented DVM behavior, typically find relatively minor vertical excursions (compared to that in temperate and tropical regions) that does not extend beneath the permanent thermocline and hence is not an important carbon export term. My takeaway from this section was that the authors were trying to leave readers with the impression that this deserves more study because it is likely substantial. In reality, the reader should be left with the impression that it is likely minor, but that there is such great uncertainty that it deserves more attention.

Lines 199-200: The authors state that in the Fe-limited SO recycling via krill and their predators is important for phytoplankton. This is true, but again, is *E. superba* special in this regard (aside from the fact that it has high biomass?). Most zooplankton play important roles in Fe-recycling and removing *E. superba* does not mean there will be no zooplankton.

Lines 209-212: After stating how important krill are for Fe-recycling, the authors state that 90% of Fe ingested by krill is egested as fecal pellets, which sink rapidly. This contradicts their previous implication that krill are important in stimulating phytoplankton production and suggests that other organisms with less rapidly sinking fecal pellets (e.g. microzooplankton) would likely lead to much greater stimulation of phytoplankton production. In other words, replacing microzooplankton will krill may decrease Fe recycling in the euphotic zone.

Lines 227-240: The authors document top-down regulation of phytoplankton by krill. This actually suggests that removing krill could lead to more phytoplankton production and potentially more carbon export. The authors never comment on the discrepancy between such results and their general assertion that krill are important in increasing the biological pump.

Lines 406-407: The authors state that the most significant uncertainty in their calculations is uncertainty in krill biomass. This is not true. In lines 399-404 they are calculating the reduction in carbon export due to removal of the carbon sequestered by krill fecal pellets using completely linear equations. The actual biomass of krill in the SO is thus irrelevant – only the amount removed actually goes into their calculations.

Lines 500 – 503: The authors suggest that krill could be included in biogeochemical models by considering them a portion of the largest zooplankton group and having that group regenerate a portion of the carbon they consume and convert another portion of what they consume to fecal pellets. It is unclear how this is in any way different from what most biogeochemical models already do with their largest zooplankton group.

Fig. 1: Given this manuscript's focus, it would make more sense if the rates and standing stocks given in this figure were focused on the SO rather than the global ocean.

I recommend that the authors reference *Euphausia superba* as *E. superba* (rather than Antarctic krill) throughout the text. This will eliminate any possibility that readers mistakenly believe that they are referring to krill living in Antarctic waters, rather than specifically to this one species, wherever it is found.

Reviewer #2 (Remarks to the Author):

In this study, Cavan et al review the role of Antarctic krill in Southern Ocean food webs and, lesser studied, in biogeochemical cycles. As a geochemist, I am by no means well placed to review the ecosystem and fisheries component of the work. However, in general, I found the concept of the review interesting, I found it was well-written, and I consider that it does offer a novel perspective suitable for publication in Nature Communications. I have predominantly only minor comments and suggestions to improve clarity, detailed below.

Comments

- The manuscript is sometimes too vague. Partially this vagueness is a result of the uncertainties in the calculations attempted, but it would be preferable to state key findings at the start of paragraphs and emphasise these in as quantitative a way as possible. For example, at line 429, the 'relatively small impact of krill fishing on the export of carbon via krill pellets' is the key finding, buried at the end of the paragraph. Be quantitative – relative to what? Yes, there are uncertainties, but this seems to be a robust finding based on Table 1.
- As a general point, as a non-expert reader I would appreciate some more context for some of the values reported. (eg lines 142-46) How do these fluxes compare to estimates for other species/physical processes?
- Overall, I felt the manuscript is longer than it needs to be. I think it can be tightened up, both by more concise, precise language (eg less 'potentially' this or that, 'relatively small') and by the shortening of some sections that seem repetitive (eg lines 82-86 – these two sentences say basically the same thing).
- I suggest rechecking some of the physical oceanography statements and references. It was my understanding that AAIW and SAMW are ventilated from surface waters originating in the Subantarctic. I appreciate that this ventilation occurs during deep winter mixing, but I don't understand how AAIW can be present at '800-1400m in the Polar Frontal Zone'. Fig. 3 is nice, but very schematic. Also, could you explain what you mean by 'carbon released into Antarctic Intermediate Water will have a longer residence time in the ocean' than that in CDW? References wise, Broecker 1991 is quite old and without a Southern Ocean focus. I suggest checking some of Lynn Talley's review papers, for example. Citing a deglacial carbon flux paper (Rose et al.) doesn't seem appropriate.

Minor comments/clarifications

Abstract: the last sentence 'Management of the Antarctic krill fishery will need to consider the combined impacts of fishing and climate change on not just krill biomass, but also on biogeochemical cycles and focus research into this important area.' Rephrase, the emphasis is odd at the end.

Line 51: move 'despite evidence of their potential importance' to line 49 after 'However,...'

Line 81: 'dominant one of'

Line 190-191: two uses of 'important'

Line 199: Reference for this statement?

Line 219 on: I struggled a bit with finding the message in this section; maybe the top down/ bottom up idea needs explaining? Or the term grazing?
Line 248: I'm not familiar with the term sympagic
Line 250-252: needs rewriting for clarity – do you mean they like the sheltered icy environment?
Line 266: Have they been partially characterised?
Line 267: What? I'm not sure what is being suggested here - that the larval faecal pellet flux might be equivalent to the respiration rate?
Line 279: Earlier you said 5-6 years in the wild? Maybe more relevant?
Line 284: Calcium concentrations are high throughout the ocean, and it's hard to imagine this process will have a major influence on P budgets? The moults will presumably break down at depths where P concs are already high"
Line 295: delete 'the time'
Line 305: AAIW
Line 417: Fragment
Line 418: First and only use of FP (faecal pellets?) acronym?
Line 434-439: A paragraph that seems a bit long winded
Line 452: shorter not smaller
Line 465: Interesting, why?
Line 474: delete comma after 'is'
Line 489-495: these statements seem to be conflicting? (Krill are not or are in food web models)
Line 499: we suggest?
Line 508: To do what?
Line 508: Our ignorance.. (remove use of 'This...')
Line 514: 'contribute to uncertainty in'
Line 522: 'regions outside of'
Line 546: 'nor has the amount of phytoplankton'
Line 547: 'potential for mixing'
Line 552: 'seasonally, and, if'
Line 583: feasibility?
Line 586: 'preserve their'
Line 588: 'Krill biomass and biogeochemical influence will both..'

Reviewer #3 (Remarks to the Author):

This article synthesizes the role and contribution of Antarctic krill in biogeochemical cycles. There has been increasing discussion about krill's potential biogeochemical role in the Southern Ocean, so having this paper to review and synthesize potentially how important krill are, is a great and very timely contribution. I am eager to see this in print. I can foresee this paper having implications to krill research, oceanography research and krill fisheries management. The figures are fantastic, especially figures 1-3. They clarify very complicated processes.

That said, I think there are a few areas where the paper needs some clarification and potential changes, including:

- The paper (e.g., starting in line 54-65) indicates it has a general focus on the role of krill writ large, but really it is focused on the role of Antarctic krill in the Southern Ocean, specifically. It would be better to be clear about this up front. The authors could then, in the end of the paper, extrapolate out to implications for krill's role across the global oceans, but the review is not of global krill. Please clarify this both up front and throughout the paper.
- In line with the above, there are instances throughout the paper where it is unclear if the authors are referring to Antarctic krill in the Southern Ocean or "krill" more generally. E.g., lines 136-146, 152-154, 158-159, 189-199, 202-215, 243-248 etc. The clarification is important so that the reader

knows if the statements refer to krill more generally or Antarctic krill specifically.

- It would also be important to state somewhere how Antarctic krill are arguably different from other krill (e.g., their large size) and the Southern Ocean of course is a very different environment than other krill environments.

- The paper is a review of Antarctic krill, yet the authors really do not draw on the research of the United States Antarctic Marine Living Resources Program, which has been conducting Antarctic krill surveys off the South Shetland Islands for more than 30 years (e.g., peer-reviewed literature by C. Reiss, J. Hinke, G. Watters, D. Kinzey). This work has contributed tremendously to the understanding of Antarctic krill population abundance, biomass and distribution, as well as age, growth and recruitment (among other life history parameters). This review paper is incomplete without including some of this work.

- The paper indicates it will touch upon the impacts of climate change on krill (e.g., lines 85-86), yet the authors do not really touch climate change, other than mentioning that we should consider it. If this is outside the scope of the study, then the authors should say that. I found that I was waiting for more content reviewing the impacts of climate change on krill and/or biogeochemical cycles in the Southern Ocean AND potential interactions with the fishing industry.

Some other specific feedback:

Line 62 and 556 and 557: Who is "we" and "our" referring to? The authors? Scientists, managers, the public? I'd replace "so we can" to "to" and I'd change "our" to "the" etc.

Line 99: Is a synoptic survey impossible or just very difficult. I would argue the latter.

Lines 149 and 451 have the statement "some (but not all)". This reads redundant. "Some" certainly implies "not all". If the "not all" papers say something interesting, then let them stand alone. Otherwise you can remove the content in parentheses.

Lines 215-217: This is a super interesting (and isolated?) example of potentially smaller scale fertilizing processes. Are there other specific examples like this? Or is this the only known one?

Lines 273-276: I would remove these lines. Again, as stated above, this review is about Antarctic krill. This was the only place another krill species was specifically mentioned and it's not clear how this relates to what happens in the Southern Ocean. If the authors want to review krill more generally, then that needs to be made clear and other species should be included.

Lines 304-305: Authors include AaIW acronym here for "Antarctic Intermediate Water" but throughout the rest of the paper they write it out in full. For clarity (many readers won't remember what AaIW refers to), I would remove the acronym and just keep it written out in full.

Lines 312-313: The authors talk about how difficult it is to quantify the role of Antarctic krill in the ecosystem. Is this true for other krill species. That is, would this task be easier or possible for a krill species that does not have such a huge (and unknown, really, as the authors point out) distribution? Do we know the biomass of other krill species in greater detail? It would give context to this paper if the authors explained Antarctic krill better in the context of global krill species (e.g., do we know more or less about *E. superba* than other krill species?).

Lines 321-356 (Box 2): There are a variety of issues here:

- Line 324: "but now rarely exceed 250,000 T/yr." Two things here. 1. Look at the overall trend since 1993; it has gone up. The historical high catches are from the USSR era. They dropped in 1993 and have been increasing since then. 2. Krill catches reached 306,145 tonnes last season (CCAMLR annual meeting report, 2018, para 5.6). Add a bar to your graph for 2018. Look at the dialogue around CCAMLR and the number of vessels intending to fish krill for the next season; it is even more than for

the 2017/18 season (<https://www.ccamlr.org/en/compliance/list-vessel-authorisations>). There is reason to believe the era of catches as low as 250,000 tonnes may be over.

- Line 327: "minimization of bycatch". Krill fisheries have to avoid seabirds and marine mammals, but have no mechanism to minimize bycatch of fish species. This has been talked about for many years at CCAMLR, including in the context of krill fisheries perhaps being the reason why some fish species (which were historically overharvested) have not recovered. The larval fish continue to be caught as bycatch in industrial krill fishing.

- Line 336: "These provide effective caps on the fishery." effective in what way?

- Line 351-352: Citation for this?

- Line 354: This line raises so many questions and does not necessarily support the statement that an increase in krill fishing remains unlikely. Global fishing reports (e.g., the "Sunken Billions") show that globally fisheries operate at a 82 billion dollar loss. Yet they continue operating and expanding where possible. If Aker is losing money, then why do they continue? Is it subsidized (and if so, then they actually might be incentivized to continue or even increase)? If they are operating at a loss, but have invested in vessels, gear and technology, then perhaps Aker (and others) would push for higher quotas so they could make a net gain. The point being that operating loss does not equate to a disincentive to fish.

- Line 355: Existing CCAMLR management measures allow for up to 620,000 tonnes taken from area 48; thus allowing for an increase from the around 250,000 tonne catch to 620,000 tonnes. A catch more than doubling in size, might qualify as "massive" to some. Further, the conservation measure for the trigger limit will come up for negotiation again in a few years. Last time it did, some countries pushed for higher limits; this may happen again and it's unclear if the trigger limit will stay.

- Lines 354-356: Per the above two points, the authors have not made a strong argument that "These economics, coupled with existing CCAMLR management measures, mean a future massive increase in the krill fishery remains unlikely." I would remove this or reword or clarify your argument.

Lines 357-431: Case Study: Antarctic krill faecal pellets and the carbon cycle. Again, a variety of issues. Overall, I think the case study adds considerable confusion and risks putting numbers on things that have too much uncertainty to really be monetized at all. I highly advise against keeping this section in the paper or at least revising it to only include the carbon estimate (not the monetization). Some explicit thoughts:

- I don't totally understand the point of this case study. The authors wording even indicates how uncertain they are: "In this section we attempt to quantify. Given all the uncertainties, which the authors point out (line 406 on), why even do this case study? What are the authors trying to show? They make the point that we really don't know krill biomass, so really don't know their importance to the carbon cycle, but then try to not only put a value on their input to the cycle but also monetize it? Both of those steps are wrought with uncertainty. If the authors publish this, some may ignore the uncertainty and instead use this study to show that krill are more valuable to the fishery than to the carbon cycle. Do the authors intend to show that that krill are more valuable when exploited? I could see value in doing a first best estimate on krill's contribution to the carbon cycle, but not monetizing it and not comparing it to the fishery.

- Line 404: It's really not clear to me where the \$338 comes from. There are no details (besides the Grant et al. citation) indicating how this was calculated.

- Line 427-429: Citation for the statement that the fishery avoids "green" krill?

Line 433: The citation #109 is a great paper, but its global in scope and does not mention the Antarctic or Southern Ocean specifically. There are so many other thorough papers that do support the authors statement more accurately (e.g. publications by KH Kock, D. Ainley and others).

Line 534-535: Citation #32 is from 2010. Has any progress been made since then towards using these new technologies to better understand krill?

Line 575: What is "they" referring to? Krill?

Line 582-584: In line with my comments on Box 2, I would reword these lines regarding the restraints on the fishery.

Line 584: Might be worth also mentioning here that not only does CCAMLR not consider biogeochemical roles of krill in management, they also do not consider climate change (or environmental change) in their decision rules (see Brooks et al. 2018 comment in Nature).

Again, this is a great paper and I hope my comments provide a means for improvement. Thanks to all the authors for this great contribution!

Reviewer #4 (Remarks to the Author):

The manuscript entitled "The importance of Antarctic krill in global biogeochemical cycles" by Cavan et al. is an excellent review about the role of krill in the fate of carbon in the Southern Ocean. They relate this importance in carbon transport to the possible impact of krill fishing on biogeochemical cycles. The manuscript put emphasis on the effect of large pelagic fauna on biogeochemistry, which is still a gap in our knowledge of the ocean. The manuscript also gives an accurate state-of-the art about the role of krill in the downward transport of carbon mainly due to krill fecal pellets (case study). Other mechanisms such as diel or seasonal vertical migrations are partially considered mainly due to the lack of knowledge about their behavior. The authors conclude the need for better estimates of krill biomass as a requisite to produce acceptable assessments of carbon transport through these large euphausiids. Finally, in general, the manuscript provides an overview of problems to solve in future studies to account for a better understanding of biogeochemical cycles and the role of krill. However, despite the good job made by the authors, I modestly suggest to slightly improve the manuscript for a better picture about the effect of krill in biogeochemical cycles.

Vertical distribution and migration. The first amendment should be related to the krill vertical migration pattern (lines 165-181). The authors are right to consider knowledge about the krill diel vertical migration quite poor. The results in the literature are quite variable and their vertical distribution rather complicated. However, there is information available about krill vertical biomass distribution in the literature (see e.g., Hernández-León et al., 2001, MEPS 223; Hernández-León et al., 2013, JMS 111) as well as respiration rates at depth (e.g., Hernández-León et al., 2008, Polar Biol. 31; Hernández-León et al., 2013). Moreover, these author found extensive diel vertical migration for small krill (down to 600 m depth) and changes in feeding during the diel cycle. These vertical migrations transport carbon downward through respiration and fecal pellet production at depth (and probably mortality there due to feeding by mesopelagic fishes). It is interesting to consider here that the vertical migration observed by Hernández-León et al. 2001 was also observed later and published in Hernández-León et al. 2013 (see their Figure 8). Curiously, the diel pattern observed was a reverse migration (downward at night). Unfortunately, this transport remains not quantified, perhaps, due to the need to replicate these results along an annual cycle.

Most important is the observation made by La et al. (2015, Estuar. Coast. Shelf Sci, 152) about the diel and seasonal vertical migration of zooplankton and nekton using acoustics (moored LADCP). They observed clear diel vertical migrations during summer and a seasonal migration in winter (lipid pump). I suggest the authors to provide this information in the review as these migrations could have an important role in the downward carbon transport due to the pelagic fauna (including krill). Unfortunately, decades of research in Antarctica did not provide the necessary data to quantify this transport but it should be a priority for future studies about biogeochemical cycles in the Southern Ocean.

Iron regeneration. In relation to the role of krill in regenerating iron the authors should also refer to the seminal paper by Tovar-Sánchez et al. (2007, Geophys. Res. Letters, 34). These authors, to my

knowledge, described for the first time the central role of krill in the iron cycling and regeneration in the Southern Ocean.

Macronutrient regeneration. The authors should also take into account the quantification and impact of ammonia excretion rates by krill measured by Lehet et al. (2012, MEPS 459). They also found a relationship between krill biomass and ammonia concentration in seawater. They concluded that besides iron regeneration by krill, their ammonia excretion also provide optimal conditions for phytoplankton growth.

Other minor problems:

Line 127: After invertebrates it is suggested to include "(mainly zooplankton)".

Line 402: State directly the price in dollars.

Lines 429-431: Include the nowadays poor quantification of downward carbon transport due to diel and seasonal (lipid pump) vertical migration.

Deliberately signed,

Santiago Hernández-León

N.B.: Sorry for the self citations in this report but I sincerely think these reverse migrations could have important biogeochemical consequences.

Response to referees

Please find below our detailed responses to the comments made by the referees on the manuscript ‘The importance of Antarctic krill in global biogeochemical cycles’ submitted to Nature Communications. We would like to take the opportunity to thank each referee for the effort they put into reviewing our manuscript, which we believe is now greatly improved.

The main elements of the manuscript we have changed, many of which were common suggestions amongst referees, are:

- Antarctic krill to *Euphausia superba* throughout the manuscript to make it clear we are mostly discussing this species, not all krill in Antarctica
- Due to concerns over the assumptions made and resulting message we have cut the Case Study section and replaced it with a new section on the different impacts that reducing *Euphausia superba* biomass could have on the Southern Ocean carbon sink. Within this section we give a shorter report of the potential decline in faecal pellet flux due to fishing, reporting this in terms of a reduction of carbon flux ($\text{mgC m}^{-2} \text{d}^{-1}$). This new section also incorporates impact of other changes in biomass (i.e. increase in krill predators and climate change), plus discussion on how removal of krill may increase some aspects of the carbon sink such as phytodetrital aggregate flux.
- We have removed emphasis on the Antarctic Intermediate Water as most krill live south of the formation zone of this water mass, and we do not know enough about how much they could contribute to the input of nutrients into this water mass. We have also removed ‘global’ from the title.
- We have edited Box 2 to acknowledge that the fishery has been increasing since the early 90s, added data for the *Euphausia superba* catch in 2018 to the figure, and removed discussion on whether or not the fishery might expand.
- We have gone into more detail on the variable diel vertical migration cycles that *Euphausia superba* undergo.
- We have removed the water mass vector from Fig. 3 and given larval krill more importance as a future research direction in Fig. 5.
- We have re-written the abstract, with most of the word count spent on summarising our biogeochemical findings with only the final sentence on the fishery.
- Even with the introduction of new text and references we have reduced the length of the manuscript by ~ 600 words and the number of references.

Specific replies to each reviewer are italicised and given below.

Reviewer #1 (Remarks to the Author):

Cavan et al. present a review of the role of *Euphausia superba* in Southern Ocean biogeochemistry. I agree with their central argument that not enough is known about the biogeochemical impact of *E. superba* (although it is important to note that this is not simply due to a lack of research, but also stems from conflicting results about, e.g., whether and/or when *E. superba* engages in DVM behaviour).

We agree that part of the difficulty in assessing the role of krill in biogeochemical cycles is due to their cryptic lifestyle. We more explicitly acknowledge this now in the section on carbon and diel vertical migrations (DVM, from line 172), where we have now addressed more of the complexities and uncertainties surrounding DVMs, which also aligns with the suggestions of Reviewer #4.

The authors give a thorough treatment of relevant studies of *E. superba*. However, they show a remarkable lack of nuance in discussing and interpreting the results of these studies. This lack of nuance runs through most of the manuscript, but is starkest in their “case study.” The authors discussion in this section basically can be summarize as: krill biomass in the region is X. Each krill produces Y amount of fecal pellet carbon per day and Z percent of these pellets sink past a depth of 100 m (or 1000 m). The flux of carbon from krill pellets is thus $F = X * Y * Z$. If 1% of these krill are removed by the fishery, carbon flux in the Southern Ocean is decreased by 1% times F. By comparing to the traded price of carbon (lines 401 – 402) it is clear that the authors are implying that catching 1% of krill biomass leads to a decrease in carbon export in the Southern Ocean by 1% * F. This ignores everything we know about how ecosystems behave and obfuscates the true issues that need to be investigated with respect to *E. superba* biogeochemical impact. Specifically, 1) the relatively short lifetimes of *E. superba* in the wild, suggest that 1% / year catch limits will not be likely to substantially change the biogeochemical impact of *E. superba* (in part because the biogeochemical impact may be limited by bottom-up food availability rather than *E. superba* abundance) and furthermore the interannual variability in *E. superba* distributions driven by natural climate cycles and/or anthropogenic forcing is certain to have a greater impact than fishing. 2) Removal of *E. superba* by fisheries does not mean that the carbon that would have been exported by fecal pellets will instead just go right back into the atmosphere. *E. superba* fecal pellet production is supported by grazing on diatoms and other organisms. If *E. superba* densities are decreased, the phytoplankton production will be shunted elsewhere in the ecosystem (and total phytoplankton production may even increase if the ecosystem is wasp-waisted as the authors suggest). Other potential fates of this phytoplankton production include (but are certainly not limited to): aggregation and direct sinking of phytoplankton (potentially greater carbon export efficiency, since no carbon is lost to respiration of krill), consumption by salps or other mesozooplankton (likely similar carbon export efficiency), or consumption by protistan zooplankton (likely reduced carbon export efficiency). I elaborate on why the authors must consider ecosystem responses in detailed comments below.

The case study was done as a ‘back-of-the-envelope’ calculation to determine how much of the pellet flux is removed by the fishery. Given your concerns which we agree with, and that of other reviewers, and the many assumptions used we have removed the case study, and replaced a vastly shorter version in a new section entitled ‘Implications of declining krill biomass on biogeochemical cycles’. The estimates we present are thus only of the reduction of the pellet flux in $\text{mgC m}^{-2} \text{d}^{-1}$ which we frame in the context of measured particulate organic carbon (POC) fluxes (lines 365-380).

Throughout this review, the authors seem to be starting with the assumption that *E. superba* plays an important role in increasing the magnitude of the biological pump and hence consistently point out the magnitude of fluxes mediated by *E. superba*. These fluxes are indeed substantial (because the biomass of *E. superba* is massive). However, the authors ignore evidence that perhaps *E. superba* actually suppresses the biological pump. Such evidence needs to be discussed. First, from a conceptual framework, it is often suggested that zooplankton grazing decreases the magnitude of the biological pump because without grazing

potentially 100% of NPP can be exported, while grazers will respire a large proportion of what they consume leading to decreased export efficiency (e.g. Buesseler & Boyd, 2009). Furthermore, while in situ sinking rate measurements do indeed find high settling velocities when krill pellets dominate flux, they can often show equally high velocities when larger aggregates dominate the flux (McDonnell & Buesseler, 2010). Furthermore, in studies that have looked at krill abundance and carbon export, there is no positive correlation between export efficiency and krill abundance (indeed, limited evidence suggests a negative correlation may be more likely, see e.g., Fig. 7B of Ducklow et al. 2006 or Fig. 19 of Stephanie Owens' MIT dissertation). Furthermore, as the authors have noted, *E. superba* abundance may actually depress total primary production, which will likely decrease the magnitude of the carbon pump.

This is a valid point that we should state the opposing impacts removing krill may have on the carbon sink. We did capture the increase of a phytodetrital sink in Fig. 3 (#2) but did not go into detail on this in the text. The new section mentioned above 'Implications of declining krill biomass on biogeochemical cycles', now addresses this by discussing how removing krill may reduce pellet and active fluxes, but could increase aggregate flux (lines 396 – 408). This section also encompasses other revised text from the first draft on the discussion of declining krill biomass due to the recovery of whale populations and climate change.

The authors devote one third of the abstract to stating that the Antarctic krill fishery must consider biogeochemical impacts of removing krill. In light of the (relatively minor) impact found by the authors' own calculations and the comments I have made above, I do not think that this sentence is justified.

Given the revised version of the manuscript, we have edited the abstract with most of the word count spent on summarising our biogeochemical findings with only the final sentence mentioning the fishery. We hope through the revised manuscript this sentence is accepted, as we now try to highlight that really there are so many unknowns, so it is difficult to quantify the exact contribution krill, and the fishery, have. But, their size and high biomass, plus physiology, means they are important for biogeochemistry and thus management should consider this.

Line 155-157: The authors state that salp fecal pellet flux attenuation in the SO is typically higher than that of krill. The studies they cite show: 1) a highly variable flux attenuation for krill pellets across many studies and 2) relatively low salp pellet attenuation in a single study. When you combine 1) the high variability in krill pellet attenuation, 2) the paucity of studies of salp pellet attenuation in the SO, 3) the general expectation that the rapidly sinking salp pellets will lead to low pellet attenuation, and 4) the repeated finding in other ecosystems that salps lead to high carbon flux to the benthos (e.g. Pfannkuche & Lochte, 1993; Smith et al. 2014), the statement that salp attenuation is "typically" higher than that of krill is unjustified. Quite simply, we don't have enough data to know what is typical.

*We have removed this part on salps. When discussing the potential of other organisms to fulfil the biogeochemical niche left by krill if *E. superba* are removed from the fishery, we have discussed that salps could possibly replace them due to their size, fast pellets and carcasses, although attenuation is variable and so more research is needed (Lines 423+).*

The authors seem to imply that *E. superba* DVM may be a substantial flux of carbon to depth. However, the results of multiple studies leave a lot of uncertainty as to whether there is a

consistent diel pattern in the species. Furthermore, the studies that have documented DVM behaviour, typically find relatively minor vertical excursions (compared to that in temperate and tropical regions) that does not extend beneath the permanent thermocline and hence is not an important carbon export term. My takeaway from this section was that the authors were trying to leave readers with the impression that this deserves more study because it is likely substantial. In reality, the reader should be left with the impression that it is likely minor, but that there is such great uncertainty that it deserves more attention.

We accept the reviewer's assessment that there remains uncertainty with respect to the DVM patterns of Antarctic krill. However, one clear pattern from a number of studies is that larval and juvenile stages can undertake extensive DVMs. As the reviewer correctly indicates, the patterns are less clear in adults, with the majority of studies showing that, even when DVM takes place, it is mainly restricted to layers above the seasonal thermocline. Therefore, whereas larvae and juveniles likely contribute to active carbon flux from respiration, adults likely do not. We have rewritten this section (lines 172+) to make these points more clearly, emphasizing the distinction between different parts of the population in terms of their respective contributions to active carbon flux.

Lines 199-200: The authors state that in the Fe-limited SO recycling via krill and their predators is important for phytoplankton. This is true, but again, is *E. superba* special in this regard (aside from the fact that it has high biomass?). Most zooplankton play important roles in Fe-recycling and removing *E. superba* does not mean there will be no zooplankton.

See response to comment below.

Lines 209-212: After stating how important krill are for Fe-recycling, the authors state that 90% of Fe ingested by krill is egested as fecal pellets, which sink rapidly. This contradicts their previous implication that krill are important in stimulating phytoplankton production and suggests that other organisms with less rapidly sinking fecal pellets (e.g. microzooplankton) would likely lead to much greater stimulation of phytoplankton production. In other words, replacing microzooplankton with krill may decrease Fe recycling in the euphotic zone.

We agree there was a contradiction in the importance of excreting Fe stimulating phytoplankton blooms, but with most iron being released in a faecal pellet. We have edited the text to state that only a small proportion is released via excretion, and that pellet Fe is not released unless the pellet is fragmented (i.e. it is not leached, line 222) which is also shown in salp pellets. However, the accumulation of iron in whales via the consumption of many krill and subsequent defecation of dissolved Fe does fuel primary production.

Lines 227-240: The authors document top-down regulation of phytoplankton by krill. This actually suggests that removing krill could lead to more phytoplankton production and potentially more carbon export. The authors never comment on the discrepancy between such results and their general assertion that krill are important in increasing the biological pump.

As mentioned above, we now discuss how removing krill could increase the biological pump in the new section 'Implications of declining krill biomass on biogeochemical cycles'.

Lines 406-407: The authors state that the most significant uncertainty in their calculations is uncertainty in krill biomass. This is not true. In lines 399-404 they are calculating the

reduction in carbon export due to removal of the carbon sequestered by krill fecal pellets using completely linear equations. The actual biomass of krill in the SO is thus irrelevant – only the amount removed actually goes into their calculations.

We have removed the case study section and now focus on the FP flux removed ($\text{mgC m}^{-2} \text{d}^{-1}$) by the fishery which is dependent on assumptions around egestion and attenuation rates.

Lines 500 – 503: The authors suggest that krill could be included in biogeochemical models by considering them a portion of the largest zooplankton group and having that group regenerate a portion of the carbon they consume and convert another portion of what they consume to fecal pellets. It is unclear how this is in any way different from what most biogeochemical models already do with their largest zooplankton group.

Not all models treat zooplankton and the release of nutrients in the same way. In many models (e.g. PISCES) zooplankton release DOM which a fraction then might be remineralised depending on a remineralisation function. However, we suggest a proportion of consumed organic matter by zooplankton should be excreted as an inorganic nutrient (e.g. NH_4), directly to the nutrient pool and not relying on remineralisation by microbes (lines 521+).

Fig. 1: Given this manuscript's focus, it would make more sense if the rates and standing stocks given in this figure were focused on the SO rather than the global ocean.

We appreciate the reviewer's stance on this, but this is not a SO-specific figure for the biological pump which would be different and include ice etc.. It is intended to inform non-biogeochemical readers about the pump generally. We also hope the figure might be used more widely by the biological pump community as it contains some processes not commonly included in similar graphics, hence we want to keep it general with global numbers. Aside from these points, we are not that the SO biological pump has been quantified sufficiently to be able to include.

I recommend that the authors reference *Euphausia superba* as *E. superba* (rather than Antarctic krill) throughout the text. This will eliminate any possibility that readers mistakenly believe that they are referring to krill living in Antarctic waters, rather than specifically to this one species, wherever it is found.

*We have changed the text to now refer to *E. superba*.*

Reviewer #2 (Remarks to the Author):

In this study, Cavan et al review the role of Antarctic krill in Southern Ocean food webs and, lesser studied, in biogeochemical cycles. As a geochemist, I am by no means well placed to review the ecosystem and fisheries component of the work. However, in general, I found the concept of the review interesting, I found it was well-written, and I consider that it does offer a novel perspective suitable for publication in Nature Communications. I have predominantly only minor comments and suggestions to improve clarity, detailed below.

Comments

- The manuscript is sometimes too vague. Partially this vagueness is a result of the uncertainties in the calculations attempted, but it would be preferable to state key findings at the start of paragraphs and emphasise these in as quantitative a way as possible. For example, at line 429, the ‘relatively small impact of krill fishing on the export of carbon via krill pellets’ is the key finding, buried at the end of the paragraph. Be quantitative – relative to what? Yes, there are uncertainties, but this seems to be a robust finding based on Table 1.

As there was much uncertainty in the case study calculations, we have removed this section and added a section ‘Implications of declining krill biomass on biogeochemical cycles’, lines 357+. We have made a succinct comparison of the decline in pellet flux due to fishing with the measured pellet fluxes in the Southern Ocean to provide a clear, quantified comparison to put the key findings in the context of other particle flux studies.

- As a general point, as a non-expert reader I would appreciate some more context for some of the values reported. (e.g. lines 142-46) How do these fluxes compare to estimates for other species/physical processes?

We have framed these values in the context of total particulate organic carbon fluxes (i.e. cells, pellets and aggregates) in other oceans.

- Overall, I felt the manuscript is longer than it needs to be. I think it can be tightened up, both by more concise, precise language (eg less ‘potentially’ this or that, ‘relatively small’) and by the shortening of some sections that seem repetitive (eg lines 82-86 – these two sentences say basically the same thing).

We have shortened the length of the manuscript by ~ 600 words.

- I suggest rechecking some of the physical oceanography statements and references. It was my understanding that AAIW and SAMW are ventilated from surface waters originating in the Subantarctic. I appreciate that this ventilation occurs during deep winter mixing, but I don’t understand how AAIW can be present at ‘800-1400m in the Polar Frontal Zone’. Fig. 3 is nice, but very schematic. Also, could you explain what you mean by ‘carbon released into Antarctic Intermediate Water will have a longer residence time in the ocean’ than that in CDW? References wise, Broecker 1991 is quite old and without a Southern Ocean focus. I suggest checking some of Lynn Talley’s review papers, for example. Citing a deglacial carbon flux paper (Rose et al.) doesn’t seem appropriate.

Thank you for the suggestion of Lynn Talley's papers. We revisited this to check and in a paper co-authored by Lynne Talley (Hartin et al 2011) the authors state:

'Circumpolar Deep Water upwells around Antarctica and is carried northward via Ekman transport as Antarctic Surface Water (AASW). AASW is converted to AAIW through air-sea fluxes equatorward of the Polar Front (PF) and subducts at the Subantarctic Front (SAF; e.g., Sloyan and Rintoul, 2001b).'

Based on this, we conclude Antarctic surface water (AASW) is formed in the ACC from the upwelling of deep circumpolar water, where some (but not most) krill live. The AASW is then subducted at the subpolar front, so the northern boundary of the ACC. The only way that krill could influence the nutrients in the AaIW is if they egest or excrete into the AASW and the nutrients are remineralised and not utilised, thus carried equatorward until subduction at the sub-polar front. In most oceans the nutrients would be utilised for photosynthesis in the surface ocean, but in the Fe-limited SO some nutrients might not get used up and be subducted. However, this is speculative with no current evidence, much of the krill population live south of the ACC and there is uncertainty on whether there would be large enough nutrient concentrations released into the AaIW to influence biogeochemical cycles elsewhere.

Therefore, we have removed 'global' from the title and the AaIW from figure 3. We now discuss in the section on transporting nutrients on krill transporting nutrients below the permanent thermocline which will sequester nutrients for > 1 year but also that type of water mass carbon and nutrients reach is important, particularly in the SO. In lines 302-306 we state other organisms living further north may be more important in these processes, in the hope of simulating research into this.

Minor comments/clarifications

All comments below have been amended in the manuscript as suggested unless stated otherwise.

Abstract: the last sentence 'Management of the Antarctic krill fishery will need to consider the combined impacts of fishing and climate change on not just krill biomass, but also on biogeochemical cycles and focus research into this important area.' Rephrase, the emphasis is odd at the end.

Line 51: move 'despite evidence of their potential importance' to line 49 after 'However,...'

Line 81: 'dominant one of'

Line 190-191: two uses of 'important'

Line 199: Reference for this statement?

Line 219 on: I struggled a bit with finding the message in this section; maybe the top down/ bottom up idea needs explaining? Or the term grazing?

This section has been re-written, lines 234+, to make the message clearer.

Line 248: I'm not familiar with the term sympagic

Replaced with 'ice-pelagic', coupling for ease of reading

Line 250-252: needs rewriting for clarity – do you mean they like the sheltered icy environment?

These sentences have been removed

Line 266: Have they been partially characterised?

Changed to 'characterised'

Line 267: What? I'm not sure what is being suggested here - that the larval faecal pellet flux might be equivalent to the respiration rate?

We removed this line

Line 279: Earlier you said 5-6 years in the wild? Maybe more relevant?

Amended to 5-6 years in the wild

Line 284: Calcium concentrations are high throughout the ocean, and it's hard to imagine this process will have a major influence on P budgets? The moults will presumably break down at depths where P concs are already high"

We give examples of Ca and P as these are elements that are in high concentrations in krill moults. We do not suggest that they will have a major influence on Ca or P budgets. This paragraph finishes with the lines 'How quickly these micro-nutrients are released from shedded exoskeletons (moults) and their possible contribution to biogeochemical cycles has yet to be quantified.', to highlight that their contribution is not known.

Line 295: delete 'the time'

Line 305: AAIW

Line 417: Fragment

This section has been removed

Line 418: First and only use of FP (faecal pellets?) acronym?

Removed

Line 434-439: A paragraph that seems a bit long winded

Section removed

Line 452: shorter not smaller

Line 465: Interesting, why?

This refers to the recent work of Ericson et al (ref 114) who found that adult krill are resilient to high CO₂ levels. We don't discuss in detail the results as we believe this is out of the remit of the manuscript, especially as we have now reduced our emphasis on climate change early on, but hope those interested will read the work of Ericson et al..

Line 474: delete comma after 'is'

Line 489-495: these statements seem to be conflicting? (Krill are not or are in food web models)

Krill are in food-web but not in biogeochemical models. This section has been edited for clarity.

Line 499: we suggest?

Line 508: To do what?

This section has been re-written to ensure clarity, 'To truly understand fully the role of a fishery or changing climate on E. superba and biogeochemical cycles we need information

on krill in today's environment but also on how they will behave and fare in a warmer more acidic ocean.'

Line 508: Our ignorance.. (remove use of 'This...')

Line 514: 'contribute to uncertainty in'

Line 522: 'regions outside of'

Line 546: 'nor has the amount of phytoplankton'

Line 547: 'potential for mixing'

Line 552: 'seasonally, and, if'

Line 583: feasibility?

Line 586: 'preserve their'

Line 588: 'Krill biomass and biogeochemical influence will both..'

Reviewer #3 (Remarks to the Author):

This article synthesizes the role and contribution of Antarctic krill in biogeochemical cycles. There has been increasing discussion about krill's potential biogeochemical role in the Southern Ocean, so having this paper to review and synthesize potentially how important krill are, is a great and very timely contribution. I am eager to see this in print. I can foresee this paper having implications to krill research, oceanography research and krill fisheries management. The figures are fantastic, especially figures 1-3. They clarify very complicated processes.

Thank you for your enthusiastic summary!

That said, I think there are a few areas where the paper needs some clarification and potential changes, including:

- The paper (e.g., starting in line 54-65) indicates it has a general focus on the role of krill writ large, but really it is focused on the role of Antarctic krill in the Southern Ocean, specifically. It would be better to be clear about this up front. The authors could then, in the end of the paper, extrapolate out to implications for krill's role across the global oceans, but the review is not of global krill. Please clarify this both up front and throughout the paper.

We now state in the Introduction more specifically that 'we focus on the role of krill (specifically Antarctic krill, Euphausia superba)' – line 68.

- In line with the above, there are instances throughout the paper where it is unclear if the authors are referring to Antarctic krill in the Southern Ocean or "krill" more generally. E.g., lines 136-146, 152-154, 158-159, 189-199, 202-215, 243-248 etc. The clarification is important so that the reader knows if the statements refer to krill more generally or Antarctic krill specifically.

We now use E. superba throughout the text to clarify when we specifically mean that species, or krill more generally.

- It would also be important to state somewhere how Antarctic krill are arguably different from other krill (e.g., their large size) and the Southern Ocean of course is a very different environment than other krill environments.

In Box 1 we state how E. superba compare to other krill species in different environments and how they are modelled differently in food-web models to other krill species. Lines 92+: 'Typically, E. superba live for 5-6 years in the wild and grow up to 65 mm in length, hence are larger than other abundant krill species some of which (e.g. Meganyctiphanes norvegica and Euphausia pacifica) play crucial roles in northern marine ecosystems. These northern species are members of much more diverse ecosystems and rarely dominate the pelagic biomass the way that E superba does. This important ecological role is reflected in the way E.superba are represented in Southern Ocean food-web models, where they are parameterised as their own functional, species-resolved group, with other Euphausiids either combined with zooplankton or as an 'Other Euphausiid' group²⁷.'

- The paper is a review of Antarctic krill, yet the authors really do not draw on the research of the United States Antarctic Marine Living Resources Program, which has been conducting Antarctic krill surveys off the South Shetland Islands for more than 30 years (e.g., peer-

reviewed literature by C. Reiss, J. Hinke, G. Watters, D. Kinzey). This work has contributed tremendously to the understanding of Antarctic krill population abundance, biomass and distribution, as well as age, growth and recruitment (among other life history parameters). This review paper is incomplete without including some of this work.

We agree the US AMLR program has hugely contributed to krill research and we now acknowledge this by referencing Reiss et al (2017, lines 183 and 187) when discussing the seasonal movement of krill and an AMLR-led study by Klein et al (2018, lines 189 and 499) on the impact of climate change on krill populations.

- The paper indicates it will touch upon the impacts of climate change on krill (e.g., lines 85-86), yet the authors do not really touch climate change, other than mentioning that we should consider it. If this is outside the scope of the study, then the authors should say that. I found that I was waiting for more content reviewing the impacts of climate change on krill and/or biogeochemical cycles in the Southern Ocean AND potential interactions with the fishing industry.

It is beyond the scope of this study to go into more detail in climate change, although it is important that it is recognised in this study. Thus, we have removed 'and also climate change' from this sentence to show the paper is focussing on the fishery.

Some other specific feedback:

All comments below have been amended in the manuscript as suggested unless stated otherwise.

Line 62 and 556 and 557: Who is “we” and “our” referring to? The authors? Scientists, managers, the public? I’d replace “so we can” to “to” and I’d change “our” to “the” etc.

Line 99: Is a synoptic survey impossible or just very difficult. I would argue the latter.

Lines 149 and 451 have the statement “some (but not all)”. This reads redundant. “Some” certainly implies “not all”. If the “not all” papers say something interesting, then let them stand alone. Otherwise you can remove the content in parentheses.

Lines 215-217: This is a super interesting (and isolated?) example of potentially smaller scale fertilizing processes. Are there other specific examples like this? Or is this the only known one?

To our knowledge this has not be shown in other regions and could be a further area of research, as removal of krill where the net effect is to enhance primary production might decrease the CO₂ drawdown and aggregate flux, but removal of krill from areas where the net effect is to decrease phytoplankton abundance would result in a decline of faecal pellet flux. We have discussed these opposing effects in the section 'Implications of declining E. Superba biomass on biogeochemical cycles', lines 396+.

Lines 273-276: I would remove these lines. Again, as stated above, this review is about Antarctic krill. This was the only place another krill species was specifically mentioned and it’s not clear how this relates to what happens in the Southern Ocean. If the authors want to

review krill more generally, then that needs to be made clear and other species should be included.

Lines 304-305: Authors include AaIW acronym here for “Antarctic Intermediate Water” but throughout the rest of the paper they write it out in full. For clarity (many readers won’t remember what AaIW refers to), I would remove the acronym and just keep it written out in full.

Lines 312-313: The authors talk about how difficult it is to quantify the role of Antarctic krill in the ecosystem. Is this true for other krill species. That is, would this task be easier or possible for a krill species that does not have such a huge (and unknown, really, as the authors point out) distribution? Do we know the biomass of other krill species in greater detail? It would give context to this paper if the authors explained Antarctic krill better in the context of global krill species (e.g., do we know more or less about *E. superba* than other krill species?).

*We have gone into more detail on *E. superba* vs. other krill in Box 1.*

Lines 321-356 (Box 2): There are a variety of issues here:

- Line 324: “but now rarely exceed 250,000 T/yr.” Two things here. 1. Look at the overall trend since 1993; it has gone up. The historical high catches are from the USSR era. They dropped in 1993 and have been increasing since then. 2. Krill catches reached 306,145 tonnes last season (CCAMLR annual meeting report, 2018, para 5.6). Add a bar to your graph for 2018. Look at the dialogue around CCAMLR and the number of vessels intending to fish krill for the next season; it is even more than for the 2017/18 season (<https://www.ccamlr.org/en/compliance/list-vessel-authorisations>). There is reason to believe the era of catches as low as 250,000 tonnes may be over.

We have added the 2018 bar. Thank you for pointing out that trend which we had not picked up on and we now address in the manuscript.

‘Catches peaked at 530,000 T yr⁻¹ in the 1980s and declined with the collapse of the Soviet fishing industry, but have since increased steadily to over 306,000 T yr⁻¹ in 2018 (Fig. Box 2)⁹¹’.

- Line 327: “minimization of bycatch”. Krill fisheries have to avoid seabirds and marine mammals, but have no mechanism to minimize bycatch of fish species. This has been talked about for many years at CCAMLR, including in the context of krill fisheries perhaps being the reason why some fish species (which were historically overharvested) have not recovered. The larval fish continue to be caught as bycatch in industrial krill fishing.

We have edited to highlight only seabirds and mammals are included in the bycatch rules.

- Line 336: “These provide effective caps on the fishery.” effective in what way?

- Line 351-352: Citation for this?

- Line 354: This line raises so many questions and does not necessarily support the statement that an increase in krill fishing remains unlikely. Global fishing reports (e.g., the “Sunken Billions”) show that globally fisheries operate at a 82 billion dollar loss. Yet they continue operating and expanding where possible. If Aker is losing money, then why do they continue? Is it subsidized (and if so, then they actually might be incentivized to continue or even increase)? If they are operating at a loss, but have invested in vessels, gear and

technology, then perhaps Aker (and others) would push for higher quotas so they could make a net gain. The point being that operating loss does not equate to a disincentive to fish.

- Line 355: Existing CCAMLR management measures allow for up to 620,000 tonnes taken from area 48; thus allowing for an increase from the around 250,000 tonne catch to 620,000 tonnes. A catch more than doubling in size, might qualify as “massive” to some. Further, the conservation measure for the trigger limit will come up for negotiation again in a few years. Last time it did, some countries pushed for higher limits; this may happen again and it’s unclear if the trigger limit will stay.

- Lines 354-356: Per the above two points, the authors have not made a strong argument that “These economics, coupled with existing CCAMLR management measures, mean a future massive increase in the krill fishery remains unlikely.” I would remove this or reword or clarify your argument.

Regarding the five comments above, we have removed text on our expectation on whether they fishery may expand or not given it is not the focus of the manuscript to make this assessment.

Lines 357-431: Case Study: Antarctic krill faecal pellets and the carbon cycle. Again, a variety of issues. Overall, I think the case study adds considerable confusion and risks putting numbers on things that have too much uncertainty to really be monetized at all. I highly advise against keeping this section in the paper or at least revising it to only include the carbon estimate (not the monetization). Some explicit thoughts:

- I don’t totally understand the point of this case study. The authors wording even indicates how uncertain they are: “In this section we attempt to quantify. Given all the uncertainties, which the authors point out (line 406 on), why even do this case study? What are the authors trying to show? They make the point that we really don’t know krill biomass, so really don’t know their importance to the carbon cycle, but then try to not only put a value on their input to the cycle but also monetize it? Both of those steps are wrought with uncertainty. If the authors publish this, some may ignore the uncertainty and instead use this study to show that krill are more valuable to the fishery than to the carbon cycle. Do the authors intend to show that that krill are more valuable when exploited? I could see value in doing a first best estimate on krill’s contribution to the carbon cycle, but not monetizing it and not comparing it to the fishery.

- Line 404: It’s really not clear to me where the \$338 comes from. There are no details (besides the Grant et al. citation) indicating how this was calculated.

- Line 427-429: Citation for the statement that the fishery avoids “green” krill?

We have now removed the case study section but retained the decline in pellet flux (in mg C) if fishing occurs at current levels. We have merged this with the section on other mechanisms on declining krill biomass (return of the whales and climate change) and also discussed the fact that removing krill may enhance the carbon sink through the increase in phytodetrital aggregates.

Line 433: The citation #109 is a great paper, but its global in scope and does not mention the Antarctic or Southern Ocean specifically. There are so many other thorough papers that do support the authors statement more accurately (e.g. publications by KH Kock, D. Ainley and others).

We have included Ainley et al (2010) on the effect of whaling on Southern Ocean food webs, lines 433 and 434.

Line 534-535: Citation #32 is from 2010. Has any progress been made since then towards using these new technologies to better understand krill?

We have added more recent references (since 2015) to support this statement.

Line 575: What is “they” referring to? Krill? *Yes, edited.*

Line 582-584: In line with my comments on Box 2, I would reword these lines regarding the restraints on the fishery.

We have removed this statement.

Line 584: Might be worth also mentioning here that not only does CCAMLR not consider biogeochemical roles of krill in management, they also do not consider climate change (or environmental change) in their decision rules (see Brooks et al. 2018 comment in Nature).

Whilst we consider this an important point to be made, we feel it is out of the scope of this paper and that our focus should remain on what CCAMLR and the scientific community can do to promote the importance of biogeochemical cycles regulated by krill.

Again, this is a great paper and I hope my comments provide a means for improvement. Thanks to all the authors for this great contribution!

Reviewer #4 (Remarks to the Author):

The manuscript entitled “The importance of Antarctic krill in global biogeochemical cycles” by Cavan et al. is an excellent review about the role of krill in the fate of carbon in the Southern Ocean. They relate this importance in carbon transport to the possible impact of krill fishing on biogeochemical cycles. The manuscript put emphasis on the effect of large pelagic fauna on biogeochemistry, which is still a gap in our knowledge of the ocean. The manuscript also gives an accurate state-of-the art about the role of krill in the downward transport of carbon mainly due to krill fecal pellets (case study). Other mechanisms such as diel or seasonal vertical migrations are partially considered mainly due to the lack of knowledge about their behaviour. The authors conclude the need for better estimates of krill biomass as a requisite to produce acceptable assessments of carbon transport through these large euphausiids. Finally, in general, the manuscript provides an overview of problems to solve in future studies to account for a better understanding of biogeochemical cycles and the role of krill. However, despite the good job made by the authors, I modestly suggest to slightly improve the manuscript for a better picture about the effect of krill in biogeochemical cycles.

Vertical distribution and migration. The first amendment should be related to the krill vertical migration pattern (lines 165-181). The authors are right to consider knowledge about the krill diel vertical migration quite poor. The results in the literature are quite variable and their vertical distribution rather complicated. However, there is information available about krill vertical biomass distribution in the literature (see e.g., Hernández-León et al., 2001, MEPS 223; Hernández-León et al., 2013, JMS 111) as well as respiration rates at depth (e.g., Hernández-León et al., 2008, Polar Biol. 31; Hernández-León et al., 2013). Moreover, these author found extensive diel vertical migration for small krill (down to 600 m depth) and changes in feeding during the diel cycle. These vertical migrations transport carbon downward through respiration and fecal pellet production at depth (and probably mortality there due to feeding by mesopelagic fishes). It is interesting to consider here that the vertical migration observed by Hernández-León et al. 2001 was also observed later and published in Hernández-León et al. 2013 (see their Figure 8). Curiously, the diel pattern observed was a reverse migration (downward at night). Unfortunately, this transport remains not quantified, perhaps, due to the need to replicate these results along an annual cycle.

We are grateful to the reviewer for their suggestions of further literature with respect to supporting our view that DVM can be substantial in certain parts of the Antarctic krill population, and we have incorporated these works into a reworking of this section lines 172+.

Most important is the observation made by La et al. (2015, Estuar. Coast. Shelf Sci, 152) about the diel and seasonal vertical migration of zooplankton and nekton using acoustics (moored LADCP). They observed clear diel vertical migrations during summer and a seasonal migration in winter (lipid pump). I suggest the authors to provide this information in the review as these migrations could have an important role in the downward carbon transport due to the pelagic fauna (including krill). Unfortunately, decades of research in Antarctica did not provide the necessary data to quantify this transport but it should be a priority for future studies about biogeochemical cycles in the Southern Ocean.

On the suggestion of the reviewer, we have mentioned more explicitly the possibility that Antarctic krill may contribute to the lipid pump as a result of a seasonal migration to deeper layers during wintertime. Although the reviewer mentions the work of La et al 2015, we note that their work believes that Euphausia crystallorophias is the most likely contributor to the acoustic patterns they resolve. We otherwise cite the works of Kane et al (2018) and Lascara et al (1999) as evidence of the tendency of Antarctic krill aggregations to be deeper during winter (lines 189+).

Iron regeneration. In relation to the role of krill in regenerating iron the authors should also refer to the seminal paper by Tovar-Sánchez et al. (2007, Geophys. Res. Letters, 34). These authors, to my knowledge, described for the first time the central role of krill in the iron cycling and regeneration in the Southern Ocean.

Thank you for bringing this article to our attention. We have now included it in the manuscript, lines 206.

Macronutrient regeneration. The authors should also take into account the quantification and impact of ammonia excretion rates by krill measured by Lehet et al. (2012, MEPS 459). They also found a relationship between krill biomass and ammonia concentration in seawater. They concluded that besides iron regeneration by krill, their ammonia excretion also provide optimal conditions for phytoplankton growth.

We have also now included this study with the rates, as well as rates around South Georgia for a comparison lines 242+.

Other minor problems:

Line 127: After invertebrates it is suggested to include “(mainly zooplankton)”.

Included

Line 402: State directly the price in dollars.

We have removed this section and the monetary value.

Lines 429-431: Include the nowadays poor quantification of downward carbon transport due to diel and seasonal (lipid pump) vertical migration.

Section and statements removed with Case Study section.

Deliberately signed,

Santiago Hernández-León

N.B.: Sorry for the self citations in this report but I sincerely think these reverse migrations could have important biogeochemical consequences.

Thank you for these suggestions.

REVIEWERS' COMMENTS:

Reviewer #1 (Remarks to the Author):

I find this version of the manuscript to be much improved and have only minor further comments:

Lines 151-154: I do not find these random estimates from different ocean basins and different depths to be useful at all. There are countless different export measurements that could have been chosen from anywhere in the world. Why highlight 3 studies for no particular reason? Comparing estimates that come from the base of the euphotic zone with estimates from 500 m depth is particularly unhelpful, because of how much of an effect flux attenuation can have in the shallow twilight zone. Why not instead give estimates specifically from the Southern Ocean (there are many) and or give estimates of the range of export values found across the planet based on global syntheses? Kanchan Maiti's Southern Ocean synthesis seems particularly useful (Maiti et al. 2013, GRL 40: 1557-1561) as does Dave Siegel's global remote sensing model, which explicitly predicts fecal pellet and total flux (Siegel et al. 2014, GBC 28, 181-196).

Line 207: Does this specifically refer to adult krill?

Lines 322 – 324: T is not a metric unit. Is this tonnes or terragrams?

Lines 509 – 513 and Table 1: This does not seem particularly useful, because it is not in any way a careful synthesis of models. It is accurate to state that fisheries-focused ECOPATH models are very biased towards higher trophic levels, but there are many other food web models (e.g. Sailley et al. 2013, MEPS 492: 253-272 or Daniels et al. 2006, DSR II 53: 532-554) that are focused on the lower food web. Why choose only PlankTOM10 out of all the biogeochemical models that have been used only in the Southern Ocean?

Lines 521 – 525: I still do not see how this is different from how large zooplankton are treated in most models. For instance, I think this is basically how large zooplankton are treated in the PlankTOM10 model that the authors cite earlier. It is also how they are treated in NEMURO and many other biogeochemical models (most biogeochemical models that have sufficient complexity to include large and small zooplankton).

Reviewer #2 (Remarks to the Author):

I am satisfied that the authors have addressed the recommendations from the first round of reviews, and that this manuscript is now acceptable for publication. Below a few very minor corrections.

Line 153: typo 'equatorial'

Line 285: Phosphorus is not a micronutrient?

Line 308-9: Ammonium is a part of the biogeochemical cycle of N

Line 363: Rephrase to 'due to the return of whales and to climate change'

Line 379: typo - a thought 'exercise' or experiment

Line 382: 'do not include'

Line 387: typo 'important'

Line 445: missing full stop after 'concentrated'

Line 470: typo 'enhance'

Line 533: chlorophyll is not a nutrient

Line 962: Should be 'DOC and DOP'? (Fig 2 caption)

Reviewer #3 (Remarks to the Author):

The manuscript looks great. Looking forward to seeing it in print.

Reviewer #4 (Remarks to the Author):

The authors addressed all the points raised by this reviewer and included the amendments proposed. I am now in favor of publishing this manuscript.

Response to referees

Our responses to the reviewers are italicised below under each reviewer point, although two of Reviewer #1 points we have addressed as one.

Reviewer #1 (Remarks to the Author):

I find this version of the manuscript to be much improved and have only minor further comments:

Lines 151-154: I do not find these random estimates from different ocean basins and different depths to be useful at all. There are countless different export measurements that could have been chosen from anywhere in the world. Why highlight 3 studies for no particular reason? Comparing estimates that come from the base of the euphotic zone with estimates from 500 m depth is particularly unhelpful, because of how much of an effect flux attenuation can have in the shallow twilight zone. Why not instead give estimates specifically from the Southern Ocean (there are many) and or give estimates of the range of export values found across the planet based on global syntheses? Kanchan Maiti's Southern Ocean synthesis seems particularly useful (Maiti et al. 2013, GRL 40: 1557-1561) as does Dave Siegel's global remote sensing model, which explicitly predicts fecal pellet and total flux (Siegel et al. 2014, GBC 28, 181-196).

We can see this point and have removed the fluxes from elsewhere in the globe. In their place as suggested by the reviewer we have used the data from Maiti et al. and to stick with the theme of the Southern Ocean also use data from Cavan et al 2015 from the Scotia Sea – which was already cited in the manuscript.

Line 207: Does this specifically refer to adult krill?

Yes, we have now specified this in the text.

Lines 322 – 324: T is not a metric unit. Is this tonnes or terragrams?

We have changed this to 't' which is the correct unit for tonnes and checked the document throughout for this.

Lines 509 – 513 and Table 1: This does not seem particularly useful, because it is not in any way a careful synthesis of models. It is accurate to state that fisheries-focused ECOPATH models are very biased towards higher trophic levels, but there are many other food web models (e.g. Sailley et al. 2013, MEPS 492: 253-272 or Daniels et al. 2006, DSR II 53: 532-554) that are focused on the lower food web. Why choose only PlankTOM10 out of all the biogeochemical models that have been used only in the Southern Ocean?

Lines 521 – 525: I still do not see how this is different from how large zooplankton are treated in most models. For instance, I think this is basically how large zooplankton are treated in the PlankTOM10 model that the authors cite earlier. It is also how they are treated in NEMURO and many other biogeochemical models (most biogeochemical models that have sufficient complexity to include large and small zooplankton).

Regarding both points above, we agree with the reviewers that fishery models poorly parameterise lower trophic levels, and that is a key point we want to make. Thus we have decided to leave in Table 1, but remove the first row with the PlankTOM10 data. Another key message from this paragraph is that having separate models for biogeochemistry and fishing is not sufficient to represent the whole system, and as a future direction we want to promote here the use of coupling these two types of models to form an end-to-end modelling approach. We have deleted the text on adapting biogeochemical models and focussed on the need for end-to-end modelling. We have cited the Sailley et al 2013 paper as recommended by the reviewer as a useful starting point for a Southern Ocean end-to-end model.

Reviewer #2 (Remarks to the Author):

I am satisfied that the authors have addressed the recommendations from the first round of reviews, and that this manuscript is now acceptable for publication. Below a few very minor corrections.

Line 153: typo 'equatorial'

Line 285: Phosphorus is not a micronutrient?

Line 308-9: Ammonium is a part of the biogeochemical cycle of N

Line 363: Rephrase to 'due to the return of whales and to climate change'

Line 379: typo - a thought 'exercise' or experiment

Line 382: 'do not include'

Line 387: typo 'important'

Line 445: missing full stop after 'concentrated'

Line 470: typo 'enhance'

Line 533: chlorophyll is not a nutrient

Line 962: Should be 'DOC and DOP'? (Fig 2 caption)

Thank you for your attention to detail. We have corrected all errors pointed out by the reviewer.

Reviewer #3 (Remarks to the Author):

The manuscript looks great. Looking forward to seeing it in print.

Thank you!

Reviewer #4 (Remarks to the Author):

The authors addressed all the points raised by this reviewer and included the amendments proposed. I am now in favor of publishing this manuscript.

Thank you!